# Variational Imbalanced Regression: Fair Uncertainty Quantification via Probabilistic Smoothing

**Ziyan Wang**
Georgia Institute of Technology
`wzy@gatech.edu`

**Hao Wang**
Rutgers University
`hw488@cs.rutgers.edu`

## Abstract

Existing regression models tend to fall short in both accuracy and uncertainty estimation when the label distribution is imbalanced. In this paper, we propose a probabilistic deep learning model, dubbed variational imbalanced regression (VIR), which not only performs well in imbalanced regression but naturally produces reasonable uncertainty estimation as a byproduct. Different from typical variational autoencoders assuming I.I.D. representations (a data point's representation is not directly affected by other data points), our VIR borrows data with similar regression labels to compute the latent representation's variational distribution; furthermore, different from deterministic regression models producing point estimates, VIR predicts the entire normal-inverse-gamma distributions and modulates the associated conjugate distributions to impose probabilistic reweighting on the imbalanced data, thereby providing better uncertainty estimation. Experiments in several real-world datasets show that our VIR can outperform state-of-the-art imbalanced regression models in terms of both accuracy and uncertainty estimation. Code will soon be available at `https://github.com/Wang-ML-Lab/variational-imbalanced-regression`.

## 1 Introduction

Deep regression models are currently the state of the art in making predictions in a continuous label space and have a wide range of successful applications in computer vision [50], natural language processing [22], healthcare [43, 45], recommender systems [16, 42], etc. However, these models fail however when the label distribution in training data is imbalanced. For example, in visual age estimation [30], where a model infers the age of a person given her visual appearance, models are typically trained on imbalanced datasets with overwhelmingly more images of younger adults, leading to poor regression accuracy for images of children or elderly people [48, 49]. Such unreliability in imbalanced regression settings motivates the need for both *improving performance for the minority* in the presence of imbalanced data and, more importantly, *providing reasonable uncertainty estimation* to inform practitioners on how reliable the predictions are (especially for the minority where accuracy is lower).

Existing methods for deep imbalanced regression (DIR) only focus on improving the accuracy of deep regression models by smoothing the label distribution and reweighting data with different labels [48, 49]. On the other hand, methods that provide uncertainty estimation for deep regression models operates under the balance-data assumption and therefore do not work well in the imbalanced setting [1, 8, 29].

To simultaneously cover these two desiderata, we propose a probabilistic deep imbalanced regression model, dubbed variational imbalanced regression (VIR). Different from typical variational autoencoders assuming I.I.D. representations (a data point's representation is not directly affected by other data points), our VIR assumes Neighboring and Identically Distributed (N.I.D.) and borrows

data with similar regression labels to compute the latent representation's variational distribution. Specifically, VIR first encodes a data point into a probabilistic representation and then mix it with neighboring representations (i.e., representations from data with similar regression labels) to produce its final probabilistic representation; VIR is therefore particularly useful for minority data as it can borrow probabilistic representations from data with similar labels (and naturally weigh them using our probabilistic model) to counteract data sparsity. Furthermore, different from deterministic regression models producing point estimates, VIR predicts the entire Normal Inverse Gamma (NIG) distributions and modulates the associated conjugate distributions by the importance weight computed from the smoothed label distribution to impose probabilistic reweighting on the imbalanced data. This allows the negative log likelihood to naturally put more focus on the minority data, thereby balancing the accuracy for data with different regression labels. Our VIR framework is compatible with any deep regression models and can be trained end to end.

We summarize our contributions as below:

- We identify the problem of probabilistic deep imbalanced regression as well as two desiderata, balanced accuracy and uncertainty estimation, for the problem.
- We propose VIR to simultaneously cover these two desiderata and achieve state-of-the-art performance compared to existing methods.
- As a byproduct, we also provide strong baselines for benchmarking high-quality uncertainty estimation and promising prediction performance on imbalanced datasets.

## 2  Related Work

**Variational Autoencoder.** Variational autoencoder (VAE) [25] is an unsupervised learning model that aims to infer probabilistic representations from data. However, as shown in Figure 1, VAE typically assumes I.I.D. representations, where a data point's representation is not directly affected by other data points. In contrast, our VIR borrows data with similar regression labels to compute the latent representation's variational distribution.

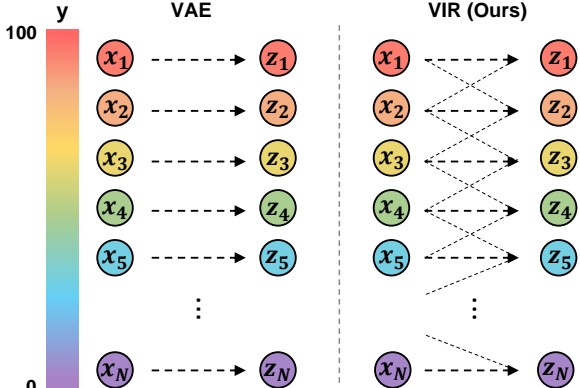

Figure 1: Comparing inference networks of typical VAE [25] and our VIR. In VAE (**left**), a data point's latent representation (i.e., $\mathbf{z}$) is affected only by itself, while in VIR (**right**), neighbors participate to modulate the final representation.

**Imbalanced Regression.** Imbalanced regression is under-explored in the machine learning community. Most existing methods for imbalanced regression are direct extensions of the SMOTE algorithm [9], a commonly used algorithm for imbalanced classification, where data from the minority classes is over-sampled. These algorithms usually synthesize augmented data for the minority regression labels by either interpolating both inputs and labels [40] or adding Gaussian noise [5, 6] (more discussion on augmentation-based methods in the Appendix).

Such algorithms fail to measure the distance in continuous label space and fall short in handling high-dimensional data (e.g., images and text). Recently, DIR [49] addresses these issues by applying kernel density estimation to smooth and reweight data on the continuous label distribution, achieving state-of-the-art performance. However, DIR only focuses on improving the accuracy, especially for the data with minority labels, and therefore does not provide uncertainty estimation, which is crucial to assess the predictions' reliability. [32] focuses on re-balancing the mean squared error (MSE) loss for imbalanced regression, and [13] introduces ranking similarity for improving deep imbalanced regression. In contrast, our VIR provides a principled probabilistic approach to simultaneously achieve these two desiderata, not only improving upon DIR in terms of performance but also producing reasonable uncertainty estimation as a much-needed byproduct to assess model reliability. There is also related work on imbalanced classification [11], which is related to our work but focusing on classification rather than regression.

**Uncertainty Estimation in Regression.** There has been renewed interest in uncertainty estimation in the context of deep regression models [1, 12, 18, 23, 26, 29, 36–38, 51]. Most existing methods directly predict the variance of the output distribution as the estimated uncertainty [1, 23, 52], rely on post-hoc confidence interval calibration [26, 37, 51], or using Bayesian neural networks [44, 46, 47]; there are also training-free approaches, such as Infer Noise and Infer Dropout [29], which produce multiple predictions from different perturbed neurons and compute their variance as uncertainty estimation. Closest to our work is Deep Evidential Regression (DER) [1], which attempts to estimate both aleatoric and epistemic uncertainty [20, 23] on regression tasks by training the neural networks to directly infer the parameters of the evidential distribution, thereby producing uncertainty measures. DER [1] is designed for the data-rich regime and therefore fails to reasonably estimate the uncertainty if the data is imbalanced; for data with minority labels, DER [1] tends produce unstable distribution parameters, leading to poor uncertainty estimation (as shown in Sec. 5). In contrast, our proposed VIR explicitly handles data imbalance in the continuous label space to avoid such instability; VIR does so by modulating both the representations and the output conjugate distribution parameters according to the imbalanced label distribution, allowing training/inference to proceed as if the data is balance and leading to better performance as well as uncertainty estimation (as shown in Sec. 5).

## 3 Method

In this section we introduce the notation and problem setting, provide an overview of our VIR, and then describe details on each of VIR's key components.

### 3.1 Notation and Problem Setting

Assuming an imbalanced dataset in continuous space $\{\mathbf{x}_i, y_i\}_{i=1}^N$ where $N$ is the total number of data points, $\mathbf{x}_i \in \mathbb{R}^d$ is the input, and $y_i \in \mathcal{Y} \subset \mathbb{R}$ is the corresponding label from a continuous label space $\mathcal{Y}$. In practice, $\mathcal{Y}$ is partitioned into B equal-interval bins $[y^{(0)}, y^{(1)}), [y^{(1)}, y^{(2)}), ..., [y^{(B-1)}, y^{(B)})$, with slight notation overload. To directly compare with baselines, we use the same grouping index for target value $b \in \mathcal{B}$ as in [49].

We denote representations as $\mathbf{z}_i$, and use $(\widetilde{\mathbf{z}}_i^\mu, \widetilde{\mathbf{z}}_i^\Sigma) = q_\phi(\mathbf{z}|\mathbf{x}_i)$ to denote the probabilistic representations for input $\mathbf{x}_i$ generated by a probabilistic encoder parameterized by $\phi$; furthermore, we denote $\bar{\mathbf{z}}$ as the mean of representation $\mathbf{z}_i$ in each bin, i.e., $\bar{\mathbf{z}} = \frac{1}{N_b} \sum_{i=1}^{N_b} \mathbf{z}_i$ for a bin with $N_b$ data points. Similarly we use $(\widehat{y}_i, \widehat{s}_i)$ to denote the mean and variance of the predictive distribution generated by a probabilistic predictor $p_\theta(y_i|\mathbf{z})$.

Figure 2: Overview of our VIR method. **Left:** The inference model infers the latent representations given input $\mathbf{x}$'s in the neighborhood. **Right:** The generative model reconstructs the input and predicts the label distribution (including the associated uncertainty) given the latent representation.

### 3.2 Method Overview

In order to achieve both desiderata in probabilistic deep imbalanced regression (i.e., performance improvement and uncertainty estimation), our proposed variational imbalanced regression (VIR) operates on both the encoder $q_\phi(\mathbf{z}_i|\{\mathbf{x}_i\}_{i=1}^N)$ and the predictor $p_\theta(y_i|\mathbf{z}_i)$.

Typical VAE [25] lower-bounds input $\mathbf{x}_i$'s marginal likelihood; in contrast, VIR lower-bounds the marginal likelihood of input $\mathbf{x}_i$ and labels $y_i$:

$$\log p_\theta(\mathbf{x}_i, y_i) = \mathcal{D}_{\mathcal{KL}}\big(q_\phi(\mathbf{z}_i|\{\mathbf{x}_i\}_{i=1}^N)||p_\theta(\mathbf{z}_i|\mathbf{x}_i, y_i)\big) + \mathcal{L}(\theta, \phi; \mathbf{x}_i, y_i).$$

Note that our variational distribution $q_\phi(\mathbf{z}_i|\{\mathbf{x}_i\}_{i=1}^N)$ (1) does not condition on labels $y_i$, since the task is to predict $y_i$ and (2) conditions on all (neighboring) inputs $\{\mathbf{x}_i\}_{i=1}^N$ rather than just $\mathbf{x}_i$. The second term $\mathcal{L}(\theta, \phi; \mathbf{x}_i, y_i)$ is VIR's evidence lower bound (ELBO), which is defined as:

$$\mathcal{L}(\theta, \phi; \mathbf{x}_i, y_i) = \underbrace{\mathbb{E}_q\big[\log p_\theta(\mathbf{x}_i|\mathbf{z}_i)\big]}_{\mathcal{L}_i^{\mathcal{P}}} + \underbrace{\mathbb{E}_q\big[\log p_\theta(y_i|\mathbf{z}_i)\big]}_{\mathcal{L}_i^{\mathcal{P}}} - \underbrace{\mathcal{D}_{\mathcal{KL}}(q_\phi(\mathbf{z}_i|\{\mathbf{x}_i\}_{i=1}^N)||p_\theta(\mathbf{z}_i))}_{\mathcal{L}_i^{\mathcal{KL}}}. \quad (1)$$

where the $p_\theta(\mathbf{z}_i)$ is the standard Gaussian prior $\mathcal{N}(\mathbf{0}, \mathbf{I})$, following typical VAE [25], and the expectation is taken over $q_\phi(\mathbf{z}_i|\{\mathbf{x}_i\}_{i=1}^N)$, which infers $\mathbf{z}_i$ by borrowing data with similar regression labels to produce the balanced probabilistic representations, which is beneficial especially for the minority (see Sec. 3.3 for details).

Different from typical regression models which produce only point estimates for $y_i$, our VIR's predictor, $p_\theta(y_i|\mathbf{z}_i)$, directly produces the parameters of the entire NIG distribution for $y_i$ and further imposes probabilistic reweighting on the imbalanced data, thereby producing balanced predictive distributions (more details in Sec. 3.4).

## 3.3 Constructing $q(\mathbf{z}_i|\{\mathbf{x}_i\}_{i=1}^N)$

To cover both desiderata, one needs to (1) produce *balanced* representations to improve performance for the data with minority labels and (2) produce *probabilistic* representations to naturally obtain reasonable uncertainty estimation for each model prediction. To learn such *balanced probabilistic representations*, we construct the encoder of our VIR (i.e., $q_\phi(\mathbf{z}_i|\{\mathbf{x}_i\}_{i=1}^N)$) by (1) first encoding a data point into a **probabilistic representation**, (2) computing **probabilistic statistics** from neighboring representations (i.e., representations from data with similar regression labels), and (3) producing the final representations via **probabilistic whitening and recoloring** using the obtained statistics.

**Intuition on Using Probabilistic Representation.** DIR uses deterministic representations, with one vector as the final representation for each data point. In contrast, our VIR uses probabilistic representations, with one vector as the mean of the representation and another vector as the variance of the representation. Such dual representation is more robust to noise and therefore leads to better prediction performance. Therefore, We first encode each data point into a probabilistic representation. Note that this is in contrast to existing work [49] that uses deterministic representations. We assume that each encoding $\mathbf{z}_i$ is a Gaussian distribution with parameters $\{\mathbf{z}_i^\mu, \mathbf{z}_i^\Sigma\}$, which are generated from the last layer in the deep neural network.

**From I.I.D. to Neighboring and Identically Distributed (N.I.D.).** Typical VAE [25] is an unsupervised learning model that aims to learn a variational representation from latent space to reconstruct the original inputs under the I.I.D. assumption; that is, in VAE, the latent value (i.e., $\mathbf{z}_i$) is generated from its own input $\mathbf{x}_i$. This I.I.D. assumption works well for data with majority labels, but significantly harms performance for data with minority labels. To address this problem, we replace the I.I.D. assumption with the N.I.D. assumption; specifically, VIR's variational latent representations still follow Gaussian distributions (i.e., $\mathcal{N}(\mathbf{z}_i^\mu, \mathbf{z}_i^\Sigma)$), but these distributions will be first calibrated using data with neighboring labels. For a data point $(\mathbf{x}_i, y_i)$ where $y_i$ is in the $b$'th bin, i.e., $y_i \in [y^{(b-1)}, y^{(b)})$, we compute $q(\mathbf{z}_i|\{\mathbf{x}_i\}_{i=1}^N) \triangleq \mathcal{N}(\mathbf{z}_i; \widetilde{\mathbf{z}}_i^\mu, \widetilde{\mathbf{z}}_i^\Sigma)$ with the following four steps.

**(1)** Mean and Covariance of Initial $\mathbf{z}_i$: $\qquad\qquad\qquad\qquad\qquad\qquad \mathbf{z}_i^\mu, \mathbf{z}_i^\Sigma = \mathcal{I}(\mathbf{x}_i),$

**(2)** Statistics of Bin $b$'s Statistics: $\qquad\qquad\qquad\qquad \boldsymbol{\mu}_b^\mu, \boldsymbol{\mu}_b^\Sigma, \boldsymbol{\Sigma}_b^\mu, \boldsymbol{\Sigma}_b^\Sigma = \mathcal{A}(\{\mathbf{z}_i^\mu, \mathbf{z}_i^\Sigma\}_{i=1}^N),$

**(3)** Smoothed Statistics of Bin $b$'s Statistics: $\qquad \widetilde{\boldsymbol{\mu}}_b^\mu, \widetilde{\boldsymbol{\mu}}_b^\Sigma, \widetilde{\boldsymbol{\Sigma}}_b^\mu, \widetilde{\boldsymbol{\Sigma}}_b^\Sigma = \mathcal{S}(\{\boldsymbol{\mu}_b^\mu, \boldsymbol{\mu}_b^\Sigma, \boldsymbol{\Sigma}_b^\mu, \boldsymbol{\Sigma}_b^\Sigma\}_{b=1}^B),$

**(4)** Mean and Covariance of Final $\mathbf{z}_i$: $\quad \widetilde{\mathbf{z}}_i^\mu, \widetilde{\mathbf{z}}_i^\Sigma = \mathcal{F}(\mathbf{z}_i^\mu, \mathbf{z}_i^\Sigma, \boldsymbol{\mu}_b^\mu, \boldsymbol{\mu}_b^\Sigma, \boldsymbol{\Sigma}_b^\mu, \boldsymbol{\Sigma}_b^\Sigma, \widetilde{\boldsymbol{\mu}}_b^\mu, \widetilde{\boldsymbol{\mu}}_b^\Sigma, \widetilde{\boldsymbol{\Sigma}}_b^\mu, \widetilde{\boldsymbol{\Sigma}}_b^\Sigma),$

where the details of functions $\mathcal{I}(\cdot)$, $\mathcal{A}(\cdot)$, $\mathcal{S}(\cdot)$, and $\mathcal{F}(\cdot)$ are described below.

**(1) Function $\mathcal{I}(\cdot)$: From Deterministic to Probabilistic Statistics.** Different from deterministic statistics in [49], our VIR's encoder uses *probabilistic statistics*, i.e., *statistics of statistics*. Specifically, VIR treats $\mathbf{z}_i$ as a distribution with the mean and covariance $(\mathbf{z}_i^\mu, \mathbf{z}_i^\Sigma) = \mathcal{I}(\mathbf{x}_i)$ rather than a deterministic vector.

As a result, all the deterministic statistics for bin $b$, $\boldsymbol{\mu}_b$, $\boldsymbol{\Sigma}_b$, $\widetilde{\boldsymbol{\mu}}_b$, and $\widetilde{\boldsymbol{\Sigma}}_b$ are replaced by distributions with the means and covariances, $(\boldsymbol{\mu}_b^\mu, \boldsymbol{\mu}_b^\Sigma)$, $(\boldsymbol{\Sigma}_b^\mu, \boldsymbol{\Sigma}_b^\Sigma)$, $(\widetilde{\boldsymbol{\mu}}_b^\mu, \widetilde{\boldsymbol{\mu}}_b^\Sigma)$, and $(\widetilde{\boldsymbol{\Sigma}}_b^\mu, \widetilde{\boldsymbol{\Sigma}}_b^\Sigma)$, respectively (more details in the following three paragraphs on $\mathcal{A}(\cdot)$, $\mathcal{S}(\cdot)$, and $\mathcal{F}(\cdot)$).

**(2) Function $\mathcal{A}(\cdot)$: Statistics of the Current Bin $b$'s Statistics.** In VIR, the *deterministic overall mean* for bin $b$ (with $N_b$ data points), $\boldsymbol{\mu}_b = \bar{\mathbf{z}} = \frac{1}{N_b} \sum_{i=1}^{N_b} \mathbf{z}_i$, becomes the *probabilistic overall mean*, i.e., a distribution of $\boldsymbol{\mu}_b$ with the mean $\boldsymbol{\mu}_b^\mu$ and covariance $\boldsymbol{\mu}_b^\Sigma$ (assuming diagonal covariance) as

follows:

$$\boldsymbol{\mu}_b^\mu \triangleq \mathbb{E}[\bar{\mathbf{z}}] = \frac{1}{N_b}\sum\nolimits_{i=1}^{N_b}\mathbb{E}[\mathbf{z}_i] = \frac{1}{N_b}\sum\nolimits_{i=1}^{N_b}\mathbf{z}_i^\mu,$$

$$\boldsymbol{\mu}_b^\Sigma \triangleq \mathbb{V}[\bar{\mathbf{z}}] = \frac{1}{N_b^2}\sum\nolimits_{i=1}^{N_b}\mathbb{V}[\mathbf{z}_i] = \frac{1}{N_b^2}\sum\nolimits_{i=1}^{N_b}\mathbf{z}_i^\Sigma.$$

Similarly, the *deterministic overall covariance* for bin $b$, $\boldsymbol{\Sigma}_b = \frac{1}{N_b}\sum_{i=1}^{N_b}(\mathbf{z}_i - \bar{\mathbf{z}})^2$, becomes the *probabilistic overall covariance*, i.e., a matrix-variate distribution [15] with the mean:

$$\boldsymbol{\Sigma}_b^\mu \triangleq \mathbb{E}[\boldsymbol{\Sigma}_b] = \frac{1}{N_b}\sum\nolimits_{i=1}^{N_b}\mathbb{E}[(\mathbf{z}_i - \bar{\mathbf{z}})^2] = \frac{1}{N_b}\sum\nolimits_{i=1}^{N_b}\left[\mathbf{z}_i^\Sigma + (\mathbf{z}_i^\mu)^2 - \left([\boldsymbol{\mu}_b^\Sigma]_i + ([\boldsymbol{\mu}_b^\mu]_i)^2\right)\right],$$

since $\mathbb{E}[\bar{\mathbf{z}}] = \boldsymbol{\mu}_b^\mu$ and $\mathbb{V}[\bar{\mathbf{z}}] = \boldsymbol{\mu}_b^\Sigma$. Note that the covariance of $\boldsymbol{\Sigma}_b$, i.e., $\boldsymbol{\Sigma}_b^\Sigma \triangleq \mathbb{V}[\boldsymbol{\Sigma}_b]$, involves computing the fourth-order moments, which is computationally prohibitive. Therefore in practice, we directly set $\boldsymbol{\Sigma}_b^\Sigma$ to zero for simplicity; empirically we observe that such simplified treatment already achieves promising performance improvement upon the state of the art. More discussions on the idea of the hierarchical structure of the statistics of statistics for smoothing are in the Appendix.

**(3) Function $\mathcal{S}(\cdot)$: Neighboring Data and Smoothed Statistics.** Next, we can borrow data from neighboring label bins $b'$ to compute the smoothed statistics of the current bin $b$ by applying a symmetric kernel $k(\cdot, \cdot)$ (e.g., Gaussian, Laplacian, and Triangular kernels). Specifically, the *probabilistic smoothed mean and covariance* are (assuming diagonal covariance):

$$\widetilde{\boldsymbol{\mu}}_b^\mu = \sum\nolimits_{b'\in\mathcal{B}}k(y_b, y_{b'})\boldsymbol{\mu}_{b'}^\mu, \quad \widetilde{\boldsymbol{\mu}}_b^\Sigma = \sum\nolimits_{b'\in\mathcal{B}}k^2(y_b, y_{b'})\boldsymbol{\mu}_{b'}^\Sigma, \quad \widetilde{\boldsymbol{\Sigma}}_b^\mu = \sum\nolimits_{b'\in\mathcal{B}}k(y_b, y_{b'})\boldsymbol{\Sigma}_{b'}.$$

**(4) Function $\mathcal{F}(\cdot)$: Probabilistic Whitening and Recoloring.** We develop a probabilistic version of the whitening and re-coloring procedure in [39, 49]. Specifically, we produce the final probabilistic representation $\{\widetilde{\mathbf{z}}_i^\mu, \widetilde{\mathbf{z}}_i^\Sigma\}$ for each data point as:

$$\widetilde{\mathbf{z}}_i^\mu = (\mathbf{z}_i^\mu - \boldsymbol{\mu}_b^\mu)\cdot\sqrt{\frac{\widetilde{\boldsymbol{\Sigma}}_b^\mu}{\boldsymbol{\Sigma}_b^\mu}} + \widetilde{\boldsymbol{\mu}}_b^\mu, \qquad \widetilde{\mathbf{z}}_i^\Sigma = (\mathbf{z}_i^\Sigma + \boldsymbol{\mu}_b^\Sigma)\cdot\sqrt{\frac{\widetilde{\boldsymbol{\Sigma}}_b^\mu}{\boldsymbol{\Sigma}_b^\mu}} + \widetilde{\boldsymbol{\mu}}_b^\Sigma. \tag{2}$$

During training, we keep updating the probabilistic overall statistics, $\{\boldsymbol{\mu}_b^\mu, \boldsymbol{\mu}_b^\Sigma, \boldsymbol{\Sigma}_b^\mu\}$, and the probabilistic smoothed statistics, $\{\widetilde{\boldsymbol{\mu}}_b^\mu, \widetilde{\boldsymbol{\mu}}_b^\Sigma, \widetilde{\boldsymbol{\Sigma}}_b^\mu\}$, across different epochs. The probabilistic representation $\{\widetilde{\mathbf{z}}_i^\mu, \widetilde{\mathbf{z}}_i^\Sigma\}$ are then re-parameterized [25] into the final representation $\mathbf{z}_i$, and passed into the final layer (discussed in Sec. 3.4) to generate the prediction and uncertainty estimation. Note that the computation of statistics from multiple $\mathbf{x}$'s is only needed during training. During testing, VIR directly uses these statistics and therefore does not need to re-compute them.

### 3.4 Constructing $p(y_i|\mathbf{z}_i)$

Our VIR's predictor $p(y_i|\mathbf{z}_i) \triangleq \mathcal{N}(y_i; \widehat{y}_i, \widehat{s}_i)$ predicts both the mean and variance for $y_i$ by first predicting the NIG distribution and then marginalizing out the latent variables. It is motivated by the following observations on label distribution smoothing (LDS) in [49] and deep evidential regression (DER) in [1], as well as intuitions on effective counts in conjugate distributions.

**LDS's Limitations in Our Probabilistic Imbalanced Regression Setting.** The motivation of LDS [49] is that the empirical label distribution can not reflect the real label distribution in an imbalanced dataset with a continuous label space; consequently, reweighting methods for imbalanced regression fail due to these inaccurate label densities. By applying a smoothing kernel on the empirical label distribution, LDS tries to recover the effective label distribution, with which reweighting methods can obtain 'better' weights to improve imbalanced regression. However, in our probabilistic imbalanced regression, one needs to consider both (1) prediction accuracy for the data with minority labels and (2) uncertainty estimation for each model. Unfortunately, LDS only focuses on improving the accuracy, especially for the data with minority labels, and therefore does not provide uncertainty estimation, which is crucial to assess the predictions' reliability.

**DER's Limitations in Our Probabilistic Imbalanced Regression Setting.** In DER [1], the predicted labels with their corresponding uncertainties are represented by the approximate posterior parameters $(\gamma, \nu, \alpha, \beta)$ of the NIG distribution $NIG(\gamma, \nu, \alpha, \beta)$. A DER model is trained via minimizing the negative log-likelihood (NLL) of a Student-t distribution:

$$\mathcal{L}_i^{DER} = \frac{1}{2}\log(\frac{\pi}{\nu}) + (\alpha + \frac{1}{2})\log((y_i - \gamma)^2\nu + \Omega) - \alpha\log(\Omega) + \log(\frac{\Gamma(\alpha)}{\Gamma(\alpha + \frac{1}{2})}), \tag{3}$$

where $\Omega = 2\beta(1 + \nu)$. It is therefore nontrivial to properly incorporate a reweighting mechanism into the NLL. One straightforward approach is to directly reweight $\mathcal{L}_i^{DER}$ for different data points $(\mathbf{x}_i, y_i)$. However, this contradicts the formulation of NIG and often leads to poor performance, as we verify in Sec. 5.

**Intuition of Pseudo-Counts for VIR.** To properly incorporate different reweighting methods, our VIR relies on the intuition of pseudo-counts (pseudo-observations) in conjugate distributions [4]. Assuming Gaussian likelihood, the *conjugate distributions* would be an NIG distribution [4], i.e., $(\mu, \Sigma) \sim NIG(\gamma, \nu, \alpha, \beta)$, which means:

$$\mu \sim \mathcal{N}(\gamma, \Sigma/\nu), \quad \Sigma \sim \Gamma^{-1}(\alpha, \beta),$$

where $\Gamma^{-1}(\alpha, \beta)$ is an inverse gamma distribution. With an NIG prior distribution $NIG(\gamma_0, \nu_0, \alpha_0, \beta_0)$, the posterior distribution of the NIG after observing $n$ real data points $\{u_i\}_{i=1}^n$ are:

$$\gamma_n = \frac{\gamma_0 \nu_0 + n\Psi}{\nu_n}, \quad \nu_n = \nu_0 + n, \quad \alpha_n = \alpha_0 + \frac{n}{2}, \quad \beta_n = \beta_0 + \frac{1}{2}(\gamma_0^2 \nu_0) + \Phi, \tag{4}$$

where $\Psi = \bar{u}$ and $\Phi = \frac{1}{2}(\sum_i u_i^2 - \gamma_n^2 \nu_n)$. Here $\nu_0$ and $\alpha_0$ can be interpreted as virtual observations, i.e., *pseudo-counts or pseudo-observations* that contribute to the posterior distribution. Overall, the mean of posterior distribution above can be interpreted as an estimation from $(2\alpha_0 + n)$ observations, with $2\alpha_0$ virtual observations and $n$ real observations. Similarly, the variance can be interpreted an estimation from $(\nu_0 + n)$ observations. This intuition is crucial in developing our VIR's predictor.

**From Pseudo-Counts to Balanced Predictive Distributions.** Based on the intuition above, we construct our predictor (i.e., $p(y_i|\mathbf{z}_i)$) by (1) generating the parameters in the posterior distribution of NIG, (2) computing re-weighted parameters by imposing the importance weights obtained from LDS, and (3) producing the final prediction with corresponding uncertainty estimation.

Based on Eqn. 4, we feed the final representation $\{\mathbf{z}_i\}_{i=1}^N$ generated from the Sec. 3.3 (Eqn. 2) into a linear layer to output the intermediate parameters $n_i, \Psi_i, \Phi_i$ for data point $(\mathbf{x}_i, y_i)$:

$$n_i, \Psi_i, \Phi_i = \mathcal{G}(\mathbf{z}_i), \quad \mathbf{z}_i \sim q(\mathbf{z}_i|\{\mathbf{x}_i\}_{i=1}^N) = \mathcal{N}(\mathbf{z}_i; \widetilde{\mathbf{z}}_i^\mu, \widetilde{\mathbf{z}}_i^\Sigma)$$

We then apply the importance weights $\left(\sum_{b' \in \mathcal{B}} k(y_b, y_{b'})p(y_{b'})\right)^{-1/2}$ calculated from the smoothed label distribution to the *pseudo-count* $n_i$ to produce the re-weighted parameters of posterior distribution of NIG, where $p(y)$ denotes the marginal distribution of $y$. Along with the pre-defined prior parameters $(\gamma_0, \nu_0, \alpha_0, \beta_0)$, we are able to compute the parameters of posterior distribution $NIG(\gamma_i, \nu_i, \alpha_i, \beta_i)$ for $(\mathbf{x}_i, y_i)$:

$$\gamma_i^* = \frac{\gamma_0 \nu_0 + \left(\sum_{b' \in \mathcal{B}} k(y_b, y_{b'})p(y_{b'})\right)^{-1/2} \cdot n_i \Psi_i}{\nu_n^*}, \quad \nu_i^* = \nu_0 + \left(\sum_{b' \in \mathcal{B}} k(y_b, y_{b'})p(y_{b'})\right)^{-1/2} \cdot n_i,$$

$$\alpha_i^* = \alpha_0 + \left(\sum_{b' \in \mathcal{B}} k(y_b, y_{b'})p(y_{b'})\right)^{-1/2} \cdot \frac{n_i}{2}, \quad \beta_i^* = \beta_0 + \frac{1}{2}(\gamma_0^2 \nu_0) + \Phi_i.$$

Based on the NIG posterior distribution, we can then compute final prediction and uncertainty estimation as

$$\widehat{y}_i = \gamma_i^*, \quad \widehat{s}_i = \frac{\beta_i^*}{\nu_i^*(\alpha_i^* - 1)}.$$

We use an objective function similar to Eqn. 3, but with different definitions of $(\gamma, \nu, \alpha, \beta)$, to optimize our VIR model:

$$\mathcal{L}_i^{\mathcal{P}} = \mathbb{E}_{q_\phi(\mathbf{z}_i|\{\mathbf{x}_i\}_{i=1}^N)} \left[\frac{1}{2}\log(\frac{\pi}{\nu_i^*}) + (\alpha_i^* + \frac{1}{2})\log((y_i - \gamma_i^*)^2 \nu_n^* + \Omega) - \alpha_i^* \log(\omega_i^*) + \log(\frac{\Gamma(\alpha_i^*)}{\Gamma(\alpha_i^* + \frac{1}{2})})\right], \tag{5}$$

where $\omega_i^* = 2\beta_i^*(1 + \nu_i^*)$. Note that $\mathcal{L}_i^{\mathcal{P}}$ is part of the ELBO in Eqn. 1. Similar to [1], we use an additional regularization term to achieve better accuracy :

$$\mathcal{L}_i^{\mathcal{R}} = (\nu + 2\alpha) \cdot |y_i - \widehat{y}_i|.$$

$\mathcal{L}_i^{\mathcal{P}}$ and $\mathcal{L}_i^{\mathcal{R}}$ together constitute the objective function for learning the predictor $p(\mathbf{y}_i|\mathbf{z}_i)$.

Table 1: Accuracy on AgeDB-DIR.

| Metrics | MAE ↓ | | | | GM ↓ | | | |
|---|---|---|---|---|---|---|---|---|
| Shot | all | many | medium | few | all | many | medium | few |
| VANILLA [49] | 7.77 | 6.62 | 9.55 | 13.67 | 5.05 | 4.23 | 7.01 | 10.75 |
| VAE [25] | 7.63 | 6.58 | 9.21 | 13.45 | 4.86 | 4.11 | 6.61 | 10.24 |
| DEEP ENS. [27] | 7.73 | 6.62 | 9.37 | 13.90 | 4.87 | 4.37 | 6.50 | 11.35 |
| INFER NOISE [29] | 8.53 | 7.62 | 9.73 | 13.82 | 5.57 | 4.95 | 6.58 | 10.86 |
| SMOTER [40] | 8.16 | 7.39 | 8.65 | 12.28 | 5.21 | 4.65 | 5.69 | 8.49 |
| SMOGN [5] | 8.26 | 7.64 | 9.01 | 12.09 | 5.36 | 4.9 | 6.19 | 8.44 |
| SQINV [49] | 7.81 | 7.16 | 8.80 | 11.2 | 4.99 | 4.57 | 5.73 | 7.77 |
| DER [1] | 8.09 | 7.31 | 8.99 | 12.66 | 5.19 | 4.59 | 6.43 | 10.49 |
| LDS [49] | 7.67 | 6.98 | 8.86 | 10.89 | 4.85 | 4.39 | 5.8 | 7.45 |
| FDS [49] | 7.69 | 7.10 | 8.86 | 9.98 | 4.83 | 4.41 | 5.97 | 6.29 |
| LDS + FDS [49] | 7.55 | 7.01 | 8.24 | 10.79 | 4.72 | 4.36 | 5.45 | 6.79 |
| RANKSIM [13] | 7.02 | 6.49 | 7.84 | 9.68 | 4.53 | 4.13 | 5.37 | 6.89 |
| LDS + FDS + DER [1] | 8.18 | 7.44 | 9.52 | 11.45 | 5.30 | 4.75 | 6.74 | 7.68 |
| VIR (OURS) | **6.99** | **6.39** | **7.47** | **9.51** | **4.41** | **4.07** | **5.05** | **6.23** |
| OURS VS. VANILLA | +0.78 | +0.23 | +2.08 | +4.16 | +0.64 | +0.16 | +1.96 | +4.52 |
| OURS VS. INFER NOISE | +1.54 | +1.23 | +2.26 | +4.31 | +1.16 | +0.88 | +1.53 | +4.63 |
| OURS VS. DER | +1.10 | +0.92 | +1.52 | +3.15 | +0.78 | +0.52 | +1.38 | +4.26 |
| OURS VS. LDS + FDS | +0.56 | +0.62 | +0.77 | +1.28 | +0.31 | +0.29 | +0.40 | +0.56 |
| OURS VS. RANKSIM | +0.03 | +0.10 | +0.37 | +0.17 | +0.12 | +0.06 | +0.32 | +0.66 |

Table 2: Accuracy on IW-DIR.

| Metrics | MAE ↓ | | | | GM ↓ | | | |
|---|---|---|---|---|---|---|---|---|
| Shot | All | Many | Medium | Few | All | Many | Medium | Few |
| VANILLA [49] | 8.06 | 7.23 | 15.12 | 26.33 | 4.57 | 4.17 | 10.59 | 20.46 |
| VAE [25] | 8.04 | 7.20 | 15.05 | 26.30 | 4.57 | 4.22 | 10.56 | 20.72 |
| DEEP ENS. [27] | 8.08 | 7.31 | 15.09 | 26.47 | 4.59 | 4.26 | 10.61 | 21.13 |
| INFER NOISE [29] | 8.11 | 7.36 | 15.23 | 26.29 | 4.68 | 4.33 | 10.65 | 20.31 |
| SMOTER [40] | 8.14 | 7.42 | 14.15 | 25.28 | 4.64 | 4.30 | 9.05 | 19.46 |
| SMOGN [5] | 8.03 | 7.30 | 14.02 | 25.93 | 4.63 | 4.30 | 8.74 | 20.12 |
| SQINV [49] | 7.87 | 7.24 | 12.44 | 22.76 | 4.47 | 4.22 | 7.25 | 15.10 |
| DER [1] | 7.85 | 7.18 | 13.35 | 24.12 | 4.47 | 4.18 | 8.18 | 15.18 |
| LDS [49] | 7.83 | 7.31 | 12.43 | 22.51 | 4.42 | 4.19 | 7.00 | 13.94 |
| FDS [49] | 7.83 | 7.23 | 12.60 | 22.37 | 4.42 | 4.20 | 6.93 | 13.48 |
| LDS + FDS [49] | 7.78 | 7.20 | 12.61 | 22.19 | 4.37 | 4.12 | 7.39 | 12.61 |
| RANKSIM [13] | 7.50 | 6.93 | 12.09 | 21.68 | 4.19 | 3.97 | 6.65 | 13.28 |
| LDS + FDS + DER [1] | 7.24 | 6.64 | 11.87 | 23.44 | 3.93 | 3.69 | 6.64 | 16.00 |
| VIR (OURS) | **7.19** | **6.56** | **11.81** | **20.96** | **3.85** | **3.63** | **6.51** | **12.23** |
| OURS VS. VANILLA | +0.87 | +0.67 | +3.31 | +5.37 | +0.72 | +0.54 | +4.08 | +8.23 |
| OURS VS. INFER NOISE | +0.92 | +0.80 | +3.42 | +5.33 | +0.83 | +0.70 | +4.14 | +8.08 |
| OURS VS. DER | +0.66 | +0.62 | +1.54 | +3.16 | +0.62 | +0.55 | +1.67 | +2.95 |
| OURS VS. LDS + FDS | +0.59 | +0.64 | +0.8 | +1.23 | +0.52 | +0.49 | +0.88 | +0.38 |
| OURS VS. RANKSIM | +0.31 | +0.37 | +0.28 | +0.72 | +0.34 | +0.34 | +0.14 | +1.05 |

## 3.5 Final Objective Function

Putting together Sec. 3.3 and Sec. 3.4, our final objective function (to minimize) for VIR is:

$$\mathcal{L}^{\mathcal{VIR}} = \sum_{i=1}^{N} \mathcal{L}_i^{\mathcal{VIR}}, \quad \mathcal{L}_i^{\mathcal{VIR}} = \lambda\mathcal{L}_i^{\mathcal{R}} - \mathcal{L}(\theta, \phi; \mathbf{x}_i, y_i) = \lambda\mathcal{L}_i^{\mathcal{R}} - \mathcal{L}_i^{\mathcal{P}} - \mathcal{L}_i^{\mathcal{D}} + \mathcal{L}_i^{\mathcal{KL}},$$

where $\mathcal{L}(\theta, \phi; \mathbf{x}_i, y_i) = \mathcal{L}_i^{\mathcal{P}} + \mathcal{L}_i^{\mathcal{D}} - \mathcal{L}_i^{\mathcal{KL}}$ is the ELBO in Eqn. 1. $\lambda$ adjusts the importance of the additional regularizer and the ELBO, and thus lead to a better result both on accuracy and uncertainty estimation.

## 4 Theory

**Notation.** As mentioned in Sec. 3.1, we partitioned $\{Y_j\}_{j=1}^{N}$ into $|\mathcal{B}|$ equal-interval bins (denote the set of bins as $\mathcal{B}$), and $\{Y_j\}_{j=1}^{N}$ are sampled from the label space $\mathcal{Y}$. In addition, We use the binary set $\{P_i\}_{i=1}^{|\mathcal{B}|}$ to represent the label distribution (frequency) for each bin $i$, i.e., $P_i \triangleq \mathbb{P}(Y_j \in \mathcal{B}_i)$. We also use the binary set $\{O_i\}_{j=1}^{N}$ to represent whether the data point $(\mathbf{x}_j, y_j)$ is observed (i.e., $O_j \sim \mathbb{P}(O_j = 1) \propto P_{B(Y_j)}$, and $\mathbb{E}_O[O_j] \sim P_{B(Y_j)}$), where $B(Y_j)$ represents the bin which $(x_j, y_j)$ belongs to. For each bin $i \in \mathcal{B}$, we denote the associated set of data points as

$$\mathcal{U}_i \triangleq \{j : Y_j = i\}.$$

When the imbalanced dataset is partially observed, we denote the observation set as:

$$\mathcal{S}_i \triangleq \{j : O_j = 1 \ \& \ B(Y_j) = i\}.$$

**Definition 4.1** (**Expectation over Observability** $\mathbb{E}_O$). *We define the expectation over the observability variable $O$ as $\mathbb{E}_O[\cdot] \equiv \mathbb{E}_{O_j \sim \mathbb{P}(O_j=1)}[\cdot]$.*

**Definition 4.2** (**True Risk**). *Based on the previous definitions, the true risk is defined as:*

$$R(\widehat{Y}) = \frac{1}{|\mathcal{B}|} \sum_{i=1}^{|\mathcal{B}|} \frac{1}{|\mathcal{U}_i|} \sum_{j \in \mathcal{U}_i} \delta_j(Y, \widehat{Y}),$$

*where $\delta_j(Y, \widehat{Y})$ refers to some loss function (e.g. MAE, MSE). In this paper, we assume the loss is upper bounded by $\Delta$, i.e., $0 \leq \delta_j(Y, \widehat{Y}) \leq \Delta$.*

Below we define the Naive Estimator.

**Definition 4.3** (**Naive Estimator**). *Given the observation set, the Naive Estimator is defined as:*

$$\widehat{R}_{\text{NAIVE}}(\widehat{Y}) = \frac{1}{\sum_{i=1}^{|\mathcal{B}|} |\mathcal{S}_i|} \sum_{i=1}^{|\mathcal{B}|} \sum_{j \in \mathcal{S}_i} \delta_j(Y, \widehat{Y}).$$

It is easy to verify that the expectation of this naive estimator is not equal to the true risk, as $\mathbb{E}_O[\widehat{R}_{\mathrm{NAIVE}}(\widehat{Y})] \neq R(\widehat{Y})$.

Considering an imbalanced dataset as a subset of observations from a balanced one, we contrast it with the Inverse Propensity Score (IPS) estimator [34].

**Definition 4.4** (**Inverse Propensity Score Estimator**). *The inverse propensity score (IPS) estimator (an unbiased estimator) is defined as*

$$\widehat{R}_{\mathrm{IPS}}(\widehat{Y}|P) = \frac{1}{|\mathcal{B}|} \sum_{i=1}^{|\mathcal{B}|} \frac{1}{|\mathcal{U}_i|} \sum_{j \in \mathcal{S}_i} \frac{\delta_j(Y, \widehat{Y})}{P_i}.$$

The IPS estimator is an unbiased estimator, as we can verify by taking the expectation value over the observation set:

$$\mathbb{E}_O[\widehat{R}_{\mathrm{IPS}}(\widehat{Y}|P)] = \frac{1}{|\mathcal{B}|} \sum_{i=1}^{|\mathcal{B}|} \frac{1}{|\mathcal{U}_i|} \sum_{j \in \mathcal{U}_i} \frac{\delta_j(Y, \widehat{Y})}{P_i} \cdot \mathbb{E}_O[O_j]$$

$$= \frac{1}{|\mathcal{B}|} \sum_{i=1}^{|\mathcal{B}|} \frac{1}{|\mathcal{U}_i|} \sum_{j \in \mathcal{U}_i} \delta_j(Y, \widehat{Y}) = R(\widehat{Y}).$$

Finally, we define our VIR/DIR estimator below.

**Definition 4.5** (**VIR Estimator**). *The VIR estimator, denoted by $\widehat{R}_{\mathrm{VIR}}(\widehat{Y}|\widetilde{P})$, is defined as:*

$$\widehat{R}_{\mathrm{VIR}}(\widehat{Y}|\widetilde{P}) = \frac{1}{|\mathcal{B}|} \sum_{i=1}^{|\mathcal{B}|} \frac{1}{|\mathcal{U}_i|} \sum_{j \in \mathcal{S}_i} \frac{\delta_j(Y, \widehat{Y})}{\widetilde{P}_i}, \tag{6}$$

*where $\{\widetilde{P}_i\}_{i=1}^{|\mathcal{B}|}$ represents the smoothed label distribution used in our VIR's objective function. It is important to note that our VIR estimator is biased.*

For multiple predictions, we select the "best" estimator according to the following definition.

**Definition 4.6** (**Empirical Risk Minimizer**). *For a given hypothesis space $\mathcal{H}$ of predictions $\widehat{Y}$, the Empirical Risk Minimization (ERM) identifies the prediction $\widehat{Y} \in \mathcal{H}$ as*

$$\widehat{Y}^{\mathrm{ERM}} = argmin_{\widehat{Y} \in \mathcal{H}} \left\{ \widehat{R}_{\mathrm{VIR}}(\widehat{Y}|\widetilde{P}) \right\}$$

With all the aforementioned definitions, we can derive the generalization bound for the VIR estimator.

**Theorem 4.1** (Generalization Bound of VIR). *In imbalanced regression with bins $\mathcal{B}$, for any finite hypothesis space of predictions $\mathcal{H} = \{\widehat{Y}_1, \ldots, \widehat{Y}_{\mathcal{H}}\}$, the transductive prediction error of the empirical risk minimizer $\widehat{Y}^{ERM}$ using the VIR estimator with estimated propensities $\widetilde{P}$ ($P_i > 0$) and given training observations $O$ from $\mathcal{Y}$ with independent Bernoulli propensities $P$, is bounded by:*

$$R(\widehat{Y}^{ERM}) \leq \widehat{R}_{\mathrm{VIR}}(\widehat{Y}^{ERM}|\widetilde{P}) + \underbrace{\frac{\Delta}{|\mathcal{B}|} \sum_{i=1}^{|\mathcal{B}|} \left| 1 - \frac{P_i}{\widetilde{P}_i} \right|}_{Bias\ Term} + \underbrace{\frac{\Delta}{|\mathcal{B}|} \sqrt{\frac{\log(2|\mathcal{H}|/\eta)}{2}} \sqrt{\sum_{i=1}^{|\mathcal{B}|} \frac{1}{\widetilde{P}_i^2}}}_{Variance\ Term}. \tag{7}$$

**Remark.** The naive estimator (i.e., Definition 4.3) has large bias and large variance. If one directly uses the original label distribution in the training objective (i.e., Definition 4.4), i.e., $\widetilde{P}_i = P_i$, the "bias" term will be 0. However, the "variance" term will be extremely large for minority data because $\widetilde{P}_i$ is very close to 0. In contrast, under VIR's N.I.D., $\widetilde{P}_i$ used in the training objective function will be smoothed. Therefore, the minority data's label density $\widetilde{P}_i$ will be smoothed out by its neighbors and becomes larger (compared to the original $P_i$), leading to smaller "variance" in the generalization error bound. Note that $\widetilde{P}_i \neq P_i$, VIR (with N.I.D.) essentially increases bias, but **significantly reduces** its variance in the imbalanced setting, thereby leading to a lower generalization error.

# 5 Experiments

**Datasets.** We evaluate our methods in terms of prediction accuracy and uncertainty estimation on four imbalanced datasets[1], AgeDB-DIR [30], IMDB-WIKI-DIR [33], STS-B-DIR [7], and NYUD2-DIR [35]. Due to page limit, results for NYUD2-DIR [35] are moved to the Appendix. We follow the preprocessing procedures in DIR [49]. Details for each datasets are in the Appendix, and please refer to [49] for details on label density distributions and levels of imbalance.

**Baselines.** We use ResNet-50 [17] (for AgeDB-DIR and IMDB-WIKI-DIR) and BiLSTM [19] (for STS-B-DIR) as our backbone networks, and more details for baseline are in the Appendix. we describe the baselines below.

- *Vanilla*: We use the term **VANILLA** to denote a plain model without adding any approaches.
- *Synthetic-Sample-Based Methods*: Various existing imbalanced regression methods are also included as baselines; these include Deep Ensemble [27], Infer Noise [29], SMOTER [40], and SMOGN [5].
- *Cost-Sensitive Reweighting*: As shown in DIR [49], the square-root weighting variant (SQINV) baseline (i.e., $\left( \sum_{b' \in \mathcal{B}} k(y_b, y_{b'}) p(y_{b'}) \right)^{-1/2}$) always outperforms Vanilla. Therefore, for fair comparison, *all* our experiments (for both baselines and VIR) use SQINV weighting.

**Evaluation Metrics.** We follow the evaluation metrics in [49] to evaluate the accuracy of our proposed methods; these include Mean Absolute Error (MAE), Mean Squared Error (MSE). Furthermore, for AgeDB-DIR and IMDB-WIKI-DIR, we use Geometric Mean (GM) to evaluate the accuracy; for STS-B-DIR, we use Pearson correlation and Spearman correlation. We use typical evaluation metrics for uncertainty estimation in regression problems to evaluate our produced uncertainty estimation; these include Negative Log Likelihood (NLL), Area Under Sparsification Error (AUSE). Eqn. 5 shows the formula for NLL, and more details regarding to AUSE can be found in [21]. We also include calibrated uncertainty results for VIR in the Appendix.

**Evaluation Process.** Following [28, 49], for a data sample $x_i$ with its label $y_i$ which falls into the target bins $b_i$, we divide the label space into three disjoint subsets: many-shot region $\{b_i \in \mathcal{B} \mid y_i \in b_i \ \& \ |y_i| > 100\}$, medium-shot region $\{b_i \in \mathcal{B} \mid y_i \in b_i \ \& \ 20 \leq |y_i| \leq 100\}$, and few-shot region $\{b_i \in \mathcal{B} \mid y_i \in b_i \ \& \ |y_i| < 20\}$, where $|\cdot|$ denotes the cardinality of the set. We report results on the overall test set and these subsets with the accuracy metrics and uncertainty metrics discussed above.

**Implementation Details.** We conducted five separate trials for our method using different random seeds. The error bars and other implementation details are included in the Appendix.

Table 3: Accuracy on STS-B-DIR.

| Metrics | MSE ↓ | | | | Pearson ↑ | | | |
|---|---|---|---|---|---|---|---|---|
| Shot | All | Many | Medium | Few | All | Many | Medium | Few |
| VANILLA [49] | 0.974 | 0.851 | 1.520 | 0.984 | 0.742 | 0.720 | 0.627 | 0.752 |
| VAE [25] | 0.968 | 0.833 | 1.511 | 1.102 | 0.751 | 0.724 | 0.621 | 0.749 |
| DEEP ENS. [27] | 0.972 | 0.846 | 1.496 | 1.032 | 0.746 | 0.723 | 0.619 | 0.750 |
| INFER NOISE [29] | 0.954 | 0.980 | 1.408 | 0.967 | 0.747 | 0.711 | 0.631 | 0.756 |
| SMOTER [40] | 1.046 | 0.924 | 1.542 | 1.154 | 0.726 | 0.693 | 0.653 | 0.706 |
| SMOGN [5] | 0.990 | 0.896 | 1.327 | 1.175 | 0.732 | 0.704 | 0.655 | 0.692 |
| INV [49] | 1.005 | 0.894 | 1.482 | 1.046 | 0.728 | 0.703 | 0.625 | 0.732 |
| DER [1] | 1.001 | 0.912 | 1.368 | 1.055 | 0.732 | 0.711 | 0.646 | 0.742 |
| LDS [49] | 0.914 | 0.819 | 1.319 | 0.955 | 0.756 | 0.734 | 0.638 | 0.762 |
| FDS [49] | 0.927 | 0.851 | 1.225 | 1.012 | 0.750 | 0.724 | 0.667 | 0.742 |
| LDS + FDS [49] | 0.907 | 0.802 | 1.363 | 0.942 | 0.760 | 0.740 | 0.652 | 0.766 |
| RANKSIM [13] | 0.903 | 0.908 | 0.911 | 0.804 | 0.758 | 0.706 | 0.690 | 0.827 |
| LDS + FDS + DER [1] | 1.007 | 0.880 | 1.535 | 1.086 | 0.729 | 0.714 | 0.635 | 0.731 |
| VIR (OURS) | **0.892** | **0.795** | **0.899** | **0.781** | **0.776** | **0.752** | **0.696** | **0.845** |
| OURS VS. VANILLA | +0.082 | +0.056 | +0.621 | +0.203 | +0.034 | +0.032 | +0.069 | +0.093 |
| OURS VS. INFER NOISE | +0.062 | +0.185 | +0.509 | +0.186 | +0.029 | +0.041 | +0.065 | +0.089 |
| OURS VS. DER | +0.109 | +0.117 | +0.469 | +0.274 | +0.044 | +0.041 | +0.050 | +0.103 |
| OURS VS. LDS + FDS | +0.015 | +0.007 | +0.464 | +0.161 | +0.016 | +0.012 | +0.044 | +0.079 |
| OURS VS. RANKSIM | +0.011 | +0.113 | +0.012 | +0.023 | +0.018 | +0.046 | +0.006 | +0.018 |

## 5.1 Imbalanced Regression Accuracy

We report the accuracy of different methods in Table 1, Table 2, and Table 3 for AgeDB-DIR, IMDB-WIKI-DIR and STS-B-DIR, respectively. In all the tables, we can observe that our VIR consistently outperforms all baselines in all metrics.

As shown in the last four rows of all three tables, our proposed VIR compares favorably against strong baselines including DIR variants [49] and DER [1], Infer Noise [29], and RankSim [13], especially on the imbalanced data samples (i.e., in the few-shot columns). Notably, VIR improves

---

[1]Among the five datasets proposed in [49], only four of them are publicly available.

Table 4: Uncertainty estimation on AgeDB-DIR.

| Metrics | NLL ↓ | | | | AUSE ↓ | | | |
|---|---|---|---|---|---|---|---|---|
| Shot | All | Many | Med | Few | All | Many | Med | Few |
| DEEP ENS. [27] | 5.311 | 4.031 | 6.726 | 8.523 | 0.541 | 0.626 | 0.466 | 0.483 |
| INFER NOISE [29] | 4.616 | 4.413 | 4.866 | 5.842 | 0.465 | 0.458 | 0.457 | 0.496 |
| DER [1] | 3.918 | 3.741 | 3.919 | 4.432 | 0.523 | 0.464 | 0.449 | 0.486 |
| LDS + FDS + DER [1] | 3.787 | 3.689 | 3.912 | 4.234 | 0.451 | 0.460 | 0.399 | 0.565 |
| **VIR (OURS)** | **3.703** | **3.598** | **3.805** | **4.196** | **0.434** | **0.456** | **0.324** | **0.414** |
| OURS VS. DER | +0.215 | +0.143 | +0.114 | +0.236 | +0.089 | +0.008 | +0.125 | +0.072 |
| OURS VS. LDS + FDS + DER | +0.084 | +0.091 | +0.107 | +0.038 | +0.017 | +0.004 | +0.075 | +0.151 |

Table 5: Uncertainty estimation on IW-DIR.

| Metrics | NLL ↓ | | | | AUSE ↓ | | | |
|---|---|---|---|---|---|---|---|---|
| Shot | All | Many | Medium | Few | All | Many | Medium | Few |
| DEEP ENS. [27] | 5.219 | 4.102 | 7.123 | 8.852 | 0.846 | 0.862 | 0.745 | 0.718 |
| INFER NOISE [29] | 4.231 | 4.078 | 5.326 | 8.292 | 0.732 | 0.728 | 0.561 | 0.478 |
| DER [1] | 3.850 | 3.699 | 4.997 | 6.638 | 0.813 | 0.802 | 0.650 | 0.541 |
| LDS + FDS + DER [1] | 3.683 | 3.602 | 4.391 | 5.697 | 0.784 | 0.670 | 0.459 | 0.483 |
| **VIR (OURS)** | **3.651** | **3.579** | **4.296** | **5.518** | **0.634** | **0.649** | **0.434** | **0.379** |
| OURS VS. DER | +0.199 | +0.120 | +0.701 | +1.120 | +0.179 | +0.153 | +0.216 | +0.162 |
| OURS VS. LDS + FDS + DER | +0.032 | +0.023 | +0.095 | +0.179 | +0.150 | +0.021 | +0.025 | +0.104 |

upon the state-of-the-art method RankSim by $9.6\%$ and $7.9\%$ on AgeDB-DIR and IMDB-WIKI-DIR, respectively, in terms of few-shot GM. This verifies the effectiveness of our methods in terms of overall performance. More accuracy results on different metrics are included in the Appendix. Besides the main results, we also include ablation studies for VIR in the Appendix, showing the effectiveness of VIR's encoder and predictor.

## 5.2 Imbalanced Regression Uncertainty Estimation

Different from DIR [49] which only focuses on accuracy, we create a new benchmark for uncertainty estimation in imbalanced regression. Table 4, Table 5, and Table 6 show the results on uncertainty estimation for three datasets AgeDB-DIR, IMDB-WIKI-DIR, and STS-B-DIR, respectively. Note that most baselines from Table 1, Table 2, and Table 3 are *deterministic* methods (as opposed to probabilistic methods like ours) and *cannot provide uncertainty estimation*; therefore are not applicable here. To show the superiority of our VIR model, we create a strongest baseline by concatenating the DIR variants (LDS + FDS) with the DER [1].

Results show that our VIR consistently outperforms all baselines across different metrics, especially in the few-shot metrics. Note that our proposed methods mainly focus on the imbalanced setting, and therefore naturally places more emphasis on the few-shot metrics. Notably, on AgeDB-DIR, IMDB-WIKI-DIR, and STS-B-DIR, our VIR improves upon the strongest baselines, by $14.2\% \sim 17.1\%$ in terms of few-shot AUSE.

## 5.3 Limitations

Although our methods successfully improve both accuracy and uncertainty estimation on imbalanced regression, there are still several limitations. Exactly computing *variance of the variances* in Sec. 3.3 is challenging; we therefore resort to fixed variance as an approximation. Developing more accurate and efficient approximations would also be interesting future work.

Table 6: Uncertainty estimation on STS-B-DIR.

| Metrics | NLL ↓ | | | | AUSE ↓ | | | |
|---|---|---|---|---|---|---|---|---|
| Shot | All | Many | Medium | Few | All | Many | Medium | Few |
| DEEP ENS. [27] | 3.913 | 3.911 | 4.223 | 4.106 | 0.709 | 0.621 | 0.676 | 0.663 |
| INFER NOISE [29] | 3.748 | 3.753 | 3.755 | 3.688 | 0.673 | 0.631 | 0.644 | 0.639 |
| DER [1] | 2.667 | 2.601 | 3.013 | 2.401 | 0.682 | 0.583 | 0.613 | 0.624 |
| LDS + FDS + DER [1] | 2.561 | 2.514 | 2.880 | 2.358 | 0.672 | 0.581 | 0.609 | 0.615 |
| **VIR (OURS)** | **1.996** | **1.810** | **2.754** | **2.152** | **0.591** | **0.575** | **0.602** | **0.510** |
| OURS VS. DER | +0.671 | +0.791 | +0.259 | +0.249 | +0.091 | +0.008 | +0.011 | +0.114 |
| OURS VS. LDS + FDS + DER | +0.565 | +0.704 | +0.126 | +0.206 | +0.081 | +0.006 | +0.007 | +0.105 |

## 6 Conclusion

We identify the problem of probabilistic deep imbalanced regression, which aims to both improve accuracy and obtain reasonable uncertainty estimation in imbalanced regression. We propose VIR, which can use any deep regression models as backbone networks. VIR borrows data with similar regression labels to produce the probabilistic representations and modulates the conjugate distributions to impose probabilistic reweighting on imbalanced data. Furthermore, we create new benchmarks with strong baselines for uncertainty estimation on imbalanced regression. Experiments show that our methods outperform state-of-the-art imbalanced regression models in terms of both accuracy and uncertainty estimation. Future work may include (1) improving VIR by better approximating *variance of the variances* in probability distributions, and (2) developing novel approaches that can achieve stable performance even on imbalanced data with limited sample size, and (3) exploring techniques such as mixture density networks [3] to enable multi-modality in the latent distribution, thereby further improving the performance.

## Acknowledgement

The authors thank Pei Wu for help with figures, the reviewers/AC for the constructive comments to improve the paper, and Amazon Web Service for providing cloud computing credit. Most work is done when ZW is a master student at UMich. HW is partially supported by NSF Grant IIS-2127918 and an Amazon Faculty Research Award. The views and conclusions contained herein are those of the authors and should not be interpreted as necessarily representing the official policies, either expressed or implied, of the sponsors.

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

# A Theory

**Notation.** As defined in Sec.3 in main paper, we partitioned $\{Y_j\}_{j=1}^N$ into $|\mathcal{B}|$ equal-interval bins (denote the set of bins as $\mathcal{B}$), and $\{Y_j\}_{j=1}^N$ are sampled from the label space $\mathcal{Y}$. In addition, We denote the binary set $\{P_i\}_{i=1}^{|\mathcal{B}|}$ as the label distribution (frequency) for each bin, i.e., $P_i \triangleq \mathbb{P}(Y_j \in \mathcal{B}_i)$. We also denote the binary set $\{O_i\}_{j=1}^N$ to represent whether the data $\{\mathbf{x}_j, y_j\}$ are observed (i.e., $O_j \sim \mathbb{P}(O_j = 1) \propto P_{B(Y_j)}$, and $\mathbb{E}_O[O_j] \sim P_{B(Y_j)}$), where $B(Y_j)$ represents the bin which $(x_j, y_j)$ belongs to. For each bin $i \in \mathcal{B}$, we denote the global set of samples as

$$\mathcal{U}_i \triangleq \{j : Y_j = i\}.$$

When the imbalanced dataset is partially observed, we denote the observation set as:

$$\mathcal{S}_i \triangleq \{j : O_j = 1 \ \& \ B(Y_j) = i\}.$$

**Definition A.1** (**Expectation over Observability** $\mathbb{E}_O$). *We define the expectation over the observability variable $O$ as $\mathbb{E}_O[\cdot] \equiv \mathbb{E}_{O_j \sim \mathbb{P}(O_j=1)}[\cdot]$*

**Definition A.2** (**True Risk**). *Based on the previous definitions, the true risk for our model is defined as:*

$$R(\widehat{Y}) = \frac{1}{|\mathcal{B}|} \sum_{i=1}^{|\mathcal{B}|} \frac{1}{|\mathcal{U}_i|} \sum_{j \in \mathcal{U}_i} \delta_j(Y, \widehat{Y}),$$

*where $\delta_j(Y, \widehat{Y})$ refers to some loss function (e.g. MAE, MSE). In this paper we assume these loss is upper bounded by $\Delta$, i.e., $0 \leq \delta_j(Y, \widehat{Y}) \leq \Delta$.*

Then in next step we define the Naive Estimator.

**Definition A.3** (**Naive Estimator**). *Given the observation set, the Naive Estimator is defined as:*

$$\widehat{R}_{\text{NAIVE}}(\widehat{Y}) = \frac{1}{\sum_{i=1}^{|\mathcal{B}|} |\mathcal{S}_i|} \sum_{i=1}^{|\mathcal{B}|} \sum_{j \in \mathcal{S}_i} \delta_j(Y, \widehat{Y})$$

*It is easy to verify that the expectation of this naive estimator is not equal to the true risk, as $\mathbb{E}_O[\widehat{R}_{\text{NAIVE}}(\widehat{Y})] \neq R(\widehat{Y})$.*

Considering an imbalanced dataset as a subset of observations from a balanced one, we contrast it with the Inverse Propensity Score (IPS) estimator [34], underscoring the superiorities of our approach.

**Definition A.4** (**Inverse Propensity Score Estimator**). *The inverse propensity score (IPS) estimator (an unbiased estimator) is defined as*

$$\widehat{R}_{\text{IPS}}(\widehat{Y}|P) = \frac{1}{|\mathcal{B}|} \sum_{i=1}^{|\mathcal{B}|} \frac{1}{|\mathcal{U}_i|} \sum_{j \in \mathcal{S}_i} \frac{\delta_j(Y, \widehat{Y})}{P_i}.$$

IPS estimator is an unbiased estimator, as we can verify by taking the expectation value over the observation set:

$$\mathbb{E}_O[\widehat{R}_{\text{IPS}}(\widehat{Y}|P)] = \frac{1}{|\mathcal{B}|} \sum_{i=1}^{|\mathcal{B}|} \frac{1}{|\mathcal{U}_i|} \sum_{j \in \mathcal{U}_i} \frac{\delta_j(Y, \widehat{Y})}{P_i} \cdot \mathbb{E}_O[O_j]$$

$$= \frac{1}{|\mathcal{B}|} \sum_{i=1}^{|\mathcal{B}|} \frac{1}{|\mathcal{U}_i|} \sum_{j \in \mathcal{U}_i} \delta_j(Y, \widehat{Y}) = R(\widehat{Y}).$$

Finally, we define our VIR/DIR estimator below.

**Definition A.5** (**VIR Estimator**). *The VIR estimator, denoted by $\widehat{R}_{\text{VIR}}(\widehat{Y}|\widetilde{P})$, is defined as:*

$$\widehat{R}_{\text{VIR}}(\widehat{Y}|\widetilde{P}) = \frac{1}{|\mathcal{B}|} \sum_{i=1}^{|\mathcal{B}|} \frac{1}{|\mathcal{U}_i|} \sum_{j \in \mathcal{S}_i} \frac{\delta_j(Y, \widehat{Y})}{\widetilde{P}_i}, \tag{8}$$

*where $\{\widetilde{P}_i\}_{i=1}^{|\mathcal{B}|}$ represents the smoothed label distribution utilized in our VIR's objective function (see Eqn.5 in the main paper). It's important to note that our VIR estimator is biased.*

For multiple predictions, we select the "best" estimator according to the following definition.

**Definition A.6** (**Empirical Risk Minimizer**). *For a given hypothesis space $\mathcal{H}$ of predictions $\widehat{Y}$, the Empirical Risk Minimization (ERM) identifies the prediction $\widehat{Y} \in \mathcal{H}$ as*

$$\widehat{Y}^{\text{ERM}} = argmin_{\widehat{Y} \in \mathcal{H}} \left\{ \widehat{R}_{\text{VIR}}(\widehat{Y}|\widetilde{P}) \right\}$$

**Lemma A.1** (**Tail Bound for VIR Estimator**). *For any given $\widehat{Y}$ and $Y$, with probability $1 - \eta$, the VIR estimator $\widehat{R}_{\text{VIR}}(\widehat{Y}|\widetilde{P})$ does not deviate from its expectation $\mathbb{E}_O[\widehat{R}_{\text{VIR}}(\widehat{Y}^{ERM}|\widetilde{P})]$ by more than*

$$\left| \widehat{R}_{\text{VIR}}(\widehat{Y}^{ERM}|\widetilde{P}) - \mathbb{E}_O[\widehat{R}_{\text{VIR}}(\widehat{Y}^{ERM}|\widetilde{P})] \right| \leq \frac{\Delta}{|\mathcal{B}|} \sqrt{\frac{\log(2|\mathcal{H}|/\eta)}{2}} \sqrt{\sum_{i=1}^{|\mathcal{B}|} \frac{1}{\widetilde{P}_i^2}}.$$

*Proof.* For independent bounded random variables $X_1, \cdots, X_n$ that takes values in intervals of sizes $\rho_1, \cdots, \rho_n$ with probability 1, and for any $\epsilon > 0$,

$$\mathbb{P}\left( \left| \sum_i^n X_i - \mathbb{E}\left[ \sum_i^n X_i \right] \right| \geq \epsilon \right) \leq 2 \exp\left( \frac{-2\epsilon^2}{\sum_i^n \rho_i^2} \right)$$

Consider the error term for each bin $i$ in $\widehat{R}_{\text{VIR}}(\widehat{Y}^{ERM}|\widetilde{P})$ as $X_i$. Using Hoeffding's inequality, define $\mathbb{P}(X_i = \frac{\delta_j(Y, \widehat{Y})}{\widetilde{P}_i}) = \widetilde{P}_i$ and $\mathbb{P}(X_i = 0) = 1 - \widetilde{P}_i$. Then, by setting $\epsilon_0 = |\mathcal{B}| \cdot \epsilon$, we are then able to show that:

$$\mathbb{P}\left( \left| |\mathcal{B}| \cdot \widehat{R}_{\text{VIR}}(\widehat{Y}^{ERM}|\widetilde{P}) - |\mathcal{B}| \cdot \mathbb{E}_O[\widehat{R}_{\text{VIR}}(\widehat{Y}^{ERM}|\widetilde{P})] \right| \geq \epsilon_0 \right) \leq 2 \exp\left( \frac{-2\epsilon_0^2}{\Delta^2 \sum_{i=1}^{|\mathcal{B}|} \frac{1}{|\mathcal{U}_i|} \sum_{j \in \mathcal{U}_i} \frac{1}{\widetilde{P}_i^2}} \right)$$

$$\iff \mathbb{P}\left( \left| \widehat{R}_{\text{VIR}}(\widehat{Y}^{ERM}|\widetilde{P}) - \mathbb{E}_O[\widehat{R}_{\text{VIR}}(\widehat{Y}^{ERM}|\widetilde{P})] \right| \geq \epsilon \right) \leq 2 \exp\left( \frac{-2\epsilon^2 |\mathcal{B}|^2}{\Delta^2 \sum_{i=1}^{|\mathcal{B}|} \frac{1}{\widetilde{P}_i^2}} \right).$$

We can then solve for $\epsilon$, completing the proof. $\qquad\square$

With all the aforementioned definitions, we can derive the generalization bound for the VIR estimator.

**Theorem A.1** (Generalization Bound of VIR). *In imbalanced regression with bins $\mathcal{B}$, for any finite hypothesis space of predictions $\mathcal{H} = \{\widehat{Y}_1, ..., \widehat{Y}_{\mathcal{H}}\}$, the transductive prediction error of the empirical risk minimizer $\widehat{Y}^{ERM}$ using the VIR estimator with estimated propensities $\widetilde{P}$ ($P_i > 0$) and given training observations $O$ from $\mathcal{Y}$ with independent Bernoulli propensities $P$, is bounded by:*

$$R(\widehat{Y}^{ERM}) \leq \widehat{R}_{\text{VIR}}(\widehat{Y}^{ERM}|\widetilde{P}) + \frac{\Delta}{|\mathcal{B}|} \sum_{i=1}^{|\mathcal{B}|} \left| 1 - \frac{P_i}{\widetilde{P}_i} \right| + \frac{\Delta}{|\mathcal{B}|} \sqrt{\frac{\log(2|\mathcal{H}|/\eta)}{2}} \sqrt{\sum_{i=1}^{|\mathcal{B}|} \frac{1}{\widetilde{P}_i^2}} \qquad (9)$$

*Proof.* We first re-state the generalization bound for our VIR estimator: with probability $1 - \eta$, we have

$$R(\widehat{Y}^{ERM}) \leq \widehat{R}_{\text{VIR}}(\widehat{Y}^{ERM}|\widetilde{P}) + \frac{\Delta}{|\mathcal{B}|} \sum_{i=1}^{|\mathcal{B}|} \left| 1 - \frac{P_i}{\widetilde{P}_i} \right| + \frac{\Delta}{|\mathcal{B}|} \sqrt{\frac{\log(2|\mathcal{H}|/\eta)}{2}} \sqrt{\sum_{i=1}^{|\mathcal{B}|} \frac{1}{\widetilde{P}_i^2}}$$

We start to prove it from the LHS:

$$R(\widehat{Y}^{ERM}) = R(\widehat{Y}^{ERM}) - \mathbb{E}_O[\widehat{R}_{\text{VIR}}(\widehat{Y}^{ERM}|\widetilde{P})] + \mathbb{E}_O[\widehat{R}_{\text{VIR}}(\widehat{Y}^{ERM}|\widetilde{P})]$$

$$= \underbrace{\text{bias}(\widehat{R}_{\text{VIR}}(\widehat{Y}^{ERM}|\widetilde{P}))}_{Bias\ Term} + \underbrace{\mathbb{E}_O[\widehat{R}_{\text{VIR}}(\widehat{Y}^{ERM}|\widetilde{P})] - \widehat{R}_{\text{VIR}}(\widehat{Y}^{ERM}|\widetilde{P})}_{Variance\ Term} + \widehat{R}_{\text{VIR}}(\widehat{Y}^{ERM}|\widetilde{P})$$

$$= \underbrace{\text{bias}(\widehat{R}_{\text{VIR}}(\widehat{Y}^{ERM}|\widetilde{P}))}_{Bias\ Term} + \left| \underbrace{\widehat{R}_{\text{VIR}}(\widehat{Y}^{ERM}|\widetilde{P}) - \mathbb{E}_O[\widehat{R}_{\text{VIR}}(\widehat{Y}^{ERM}|\widetilde{P})]}_{Variance\ Term} \right| + \widehat{R}_{\text{VIR}}(\widehat{Y}^{ERM}|\widetilde{P})$$

Below we derive each term:

***Variance Term.*** With probability $1 - \eta$, the variance term is derived as

$$\mathbb{P}\left(\left|\widehat{R}_{\text{VIR}}(\widehat{Y}^{ERM}|\widetilde{P}) - \mathbb{E}_O[\widehat{R}_{\text{VIR}}(\widehat{Y}^{ERM}|\widetilde{P})]\right| \le \epsilon\right) \ge 1 - \eta$$

$$\Longleftarrow \mathbb{P}\left(\max_{\widehat{Y}_j}\left|\widehat{R}_{\text{VIR}}(\widehat{Y}|\widetilde{P}) - \mathbb{E}_O[\widehat{R}_{\text{VIR}}(\widehat{Y}|\widetilde{P})]\right| \le \epsilon\right) \ge 1 - \eta$$

$$\Longleftrightarrow \mathbb{P}\left(\bigvee_{\widehat{Y}_j}\left|\widehat{R}_{\text{VIR}}(\widehat{Y}|\widetilde{P}) - \mathbb{E}_O[\widehat{R}_{\text{VIR}}(\widehat{Y}|\widetilde{P})]\right| \ge \epsilon\right) < \eta$$

$$\Longleftrightarrow \mathbb{P}\left(\bigcup_{\widehat{Y}_j}\left|\widehat{R}_{\text{VIR}}(\widehat{Y}|\widetilde{P}) - \mathbb{E}_O[\widehat{R}_{\text{VIR}}(\widehat{Y}|\widetilde{P})]\right| \ge \epsilon\right) < \eta$$

$$\Longleftarrow \sum_j^{|\mathcal{H}|}\mathbb{P}\left(\left|\widehat{R}_{\text{VIR}}(\widehat{Y}|\widetilde{P}) - \mathbb{E}_O[\widehat{R}_{\text{VIR}}(\widehat{Y}|\widetilde{P})]\right| \ge \epsilon\right) < \eta \tag{10}$$

$$\Longleftarrow |\mathcal{H}| \cdot 2\exp\left(\frac{-2\epsilon^2|\mathcal{B}|^2}{\Delta^2 \sum_{i=1}^{|\mathcal{B}|}\frac{1}{|\mathcal{U}_i|}\sum_{j\in\mathcal{U}_i}\frac{1}{\widetilde{P}_i^2}}\right) < \eta \tag{11}$$

$$\Longleftrightarrow |\mathcal{H}| \cdot 2\exp\left(\frac{-2\epsilon^2|\mathcal{B}|^2}{\Delta^2 \sum_{i=1}^{|\mathcal{B}|}\frac{1}{\widetilde{P}_i^2}}\right) < \eta,$$

where inequality (10) is by Boole's inequality (Union bound), and inequality (11) holds by Lemma A.1. Then, by solving for $\epsilon$, we can derive Variance Term that with probability $1 - \eta$,

$$\mathbb{E}_O[\widehat{R}_{\text{VIR}}(\widehat{Y}^{ERM}|\widetilde{P})] - \widehat{R}_{\text{VIR}}(\widehat{Y}^{ERM}|\widetilde{P}) \le \left|\widehat{R}_{\text{VIR}}(\widehat{Y}^{ERM}|\widetilde{P}) - \mathbb{E}_O[\widehat{R}_{\text{VIR}}(\widehat{Y}^{ERM}|\widetilde{P})]\right|$$

$$\le \frac{\Delta}{|\mathcal{B}|}\sqrt{\frac{\log(2|\mathcal{H}|/\eta)}{2}}\sqrt{\sum_{i=1}^{|\mathcal{B}|}\frac{1}{\widetilde{P}_i^2}}.$$

***Bias Term.*** By definition, we can derive:

$$\text{bias}(\widehat{R}_{\text{VIR}}(\widehat{Y}^{ERM}|\widetilde{P})) = R(\widehat{Y}^{ERM}) - \mathbb{E}_O[\widehat{R}_{\text{VIR}}(\widehat{Y}^{ERM}|\widetilde{P})]$$

$$= \frac{1}{|\mathcal{B}|}\sum_{i=1}^{|\mathcal{B}|}\frac{1}{|\mathcal{U}_i|}\sum_{j\in\mathcal{U}_i}\delta_j(Y,\widehat{Y}^{ERM}) - \frac{1}{|\mathcal{B}|}\sum_{i=1}^{|\mathcal{B}|}\frac{1}{|\mathcal{U}_i|}\sum_{j\in\mathcal{U}_i}\frac{P_i}{\widetilde{P}_i}\delta_j(Y,\widehat{Y}^{ERM})$$

$$\le \frac{\Delta}{|\mathcal{B}|}\sum_{i=1}^{|\mathcal{B}|}\frac{1}{|\mathcal{U}_i|}\sum_{j\in\mathcal{U}_i}\left|1 - \frac{P_i}{\widetilde{P}_i}\right|$$

$$= \frac{\Delta}{|\mathcal{B}|}\sum_{i=1}^{|\mathcal{B}|}\left|1 - \frac{P_i}{\widetilde{P}_i}\right|,$$

concluding the proof for the Bias Term, hence completing the proof for the whole generalization bound. $\qquad\square$

## B  Details for Experiments

**Datasets.** In this work, we evaluate our methods in terms of prediction accuracy and uncertainty estimation on four imbalanced datasets[2], AgeDB [30], IMDB-WIKI [33], STS-B [7], and NYUD2-DIR [35]. Due to page limit, the results for NYUD2-DIR [35] are in the supplementary. We follow

---

[2]Among the five datasets proposed in [49], only four of them are publicly available.

Table 7: **Accuracy on AgeDB-DIR.** For baselines, we directly use the reported performance in their paper and therefore do not have error bars.

| Metrics | MSE↓ | | | | MAE↓ | | | | GM↓ | | | |
|---|---|---|---|---|---|---|---|---|---|---|---|---|
| Shot | All | Many | Medium | Few | All | Many | Medium | Few | All | Many | Medium | Few |
| VANILLA [49] | 101.60 | 78.40 | 138.52 | 253.74 | 7.77 | 6.62 | 9.55 | 13.67 | 5.05 | 4.23 | 7.01 | 10.75 |
| VAE [25] | 99.85 | 78.86 | 130.59 | 223.09 | 7.63 | 6.58 | 9.21 | 13.45 | 4.86 | 4.11 | 6.61 | 10.24 |
| DEEP ENSEMBLE [27] | 100.94 | 79.3 | 129.95 | 249.18 | 7.73 | 6.62 | 9.37 | 13.90 | 4.87 | 4.37 | 6.50 | 11.35 |
| INFER NOISE [29] | 119.46 | 95.02 | 149.84 | 266.29 | 8.53 | 7.62 | 9.73 | 13.82 | 5.57 | 4.95 | 6.58 | 10.86 |
| SMOTER [40] | 114.34 | 93.35 | 129.89 | 244.57 | 8.16 | 7.39 | 8.65 | 12.28 | 5.21 | 4.65 | 5.69 | 8.49 |
| SMOGN [5] | 117.29 | 101.36 | 133.86 | 232.90 | 8.26 | 7.64 | 9.01 | 12.09 | 5.36 | 4.9 | 6.19 | 8.44 |
| SQINV [49] | 105.14 | 87.21 | 127.66 | 212.30 | 7.81 | 7.16 | 8.80 | 11.2 | 4.99 | 4.57 | 5.73 | 7.77 |
| DER [1] | 106.77 | 91.29 | 122.43 | 209.69 | 8.09 | 7.31 | 8.99 | 12.66 | 5.19 | 4.59 | 6.43 | 10.49 |
| LDS [49] | 102.22 | 83.62 | 128.73 | 204.64 | 7.67 | 6.98 | 8.86 | 10.89 | 4.85 | 4.39 | 5.8 | 7.45 |
| FDS [49] | 101.67 | 86.49 | 129.61 | 167.75 | 7.69 | 7.10 | 8.86 | 9.98 | 4.83 | 4.41 | 5.97 | 6.29 |
| LDS + FDS [49] | 99.46 | 84.10 | 112.20 | 209.27 | 7.55 | 7.01 | 8.24 | 10.79 | 4.72 | 4.36 | 5.45 | 6.79 |
| RANKSIM [13] | 83.51 | 71.99 | 99.14 | 149.05 | 7.02 | 6.49 | 7.84 | 9.68 | 4.53 | 4.13 | 5.37 | 6.89 |
| LDS + FDS + DER [1] | 112.62 | 94.21 | 140.03 | 210.72 | 8.18 | 7.44 | 9.52 | 11.45 | 5.30 | 4.75 | 6.74 | 7.68 |
| VIR (OURS) | 81.76±0.10 | 70.61±0.05 | 91.47±1.50 | 142.36±2.10 | 6.99±0.02 | 6.39±0.02 | 7.47±0.04 | 9.51±0.06 | 4.41±0.03 | 4.07±0.02 | 5.05±0.03 | 6.23±0.05 |
| OURS VS. VANILLA | +19.84 | +7.79 | +47.05 | +111.38 | +0.78 | +0.23 | +2.08 | +4.16 | +0.64 | +0.16 | +1.96 | +4.52 |
| OURS VS. INFER NOISE | +37.70 | +24.41 | +58.37 | +123.93 | +1.54 | +1.23 | +2.26 | +4.31 | +1.16 | +0.88 | +1.53 | +4.63 |
| OURS VS. DER | +25.01 | +20.68 | +30.96 | +67.33 | +1.10 | +0.92 | +1.52 | +3.15 | +0.78 | +0.52 | +1.38 | +4.26 |
| OURS VS. LDS + FDS | +17.70 | +13.49 | +20.73 | +66.91 | +0.56 | +0.62 | +0.77 | +1.28 | +0.31 | +0.29 | +0.40 | +0.56 |
| OURS VS. RANKSIM | +1.75 | +1.38 | +7.67 | +6.69 | +0.03 | +0.10 | +0.37 | +0.17 | +0.12 | +0.06 | +0.32 | +0.66 |

Table 8: **Accuracy on IMDB-WIKI-DIR.** For baselines, we directly use the reported performance in their paper and therefore do not have error bars.

| Metrics | MSE↓ | | | | MAE↓ | | | | GM↓ | | | |
|---|---|---|---|---|---|---|---|---|---|---|---|---|
| Shot | All | Many | Medium | Few | All | Many | Medium | Few | All | Many | Medium | Few |
| VANILLA [49] | 138.06 | 108.70 | 366.09 | 964.92 | 8.06 | 7.23 | 15.12 | 26.33 | 4.57 | 4.17 | 10.59 | 20.46 |
| VAE [25] | 137.98 | 108.62 | 361.74 | 964.87 | 8.04 | 7.20 | 15.05 | 26.30 | 4.57 | 4.22 | 10.56 | 20.72 |
| DEEP ENSEMBLE [27] | 138.02 | 108.83 | 365.76 | 962.88 | 8.08 | 7.31 | 15.09 | 26.47 | 4.59 | 4.26 | 10.61 | 21.13 |
| INFER NOISE [29] | 143.62 | 112.26 | 373.19 | 961.97 | 8.11 | 7.36 | 15.23 | 26.29 | 4.68 | 4.33 | 10.65 | 20.31 |
| SMOTER [40] | 138.75 | 111.55 | 346.09 | 935.89 | 8.14 | 7.42 | 14.15 | 25.28 | 4.64 | 4.30 | 9.05 | 19.46 |
| SMOGN [5] | 136.09 | 109.15 | 339.09 | 944.20 | 8.03 | 7.30 | 14.02 | 25.93 | 4.63 | 4.30 | 8.74 | 20.12 |
| SQINV [49] | 134.36 | 111.23 | 308.63 | 834.08 | 7.87 | 7.24 | 12.44 | 22.76 | 4.47 | 4.22 | 7.25 | 15.10 |
| DER [1] | 133.81 | 107.51 | 332.90 | 916.18 | 7.85 | 7.18 | 13.35 | 24.12 | 4.47 | 4.18 | 8.18 | 15.18 |
| LDS [49] | 131.65 | 109.04 | 298.98 | 829.35 | 7.83 | 7.31 | 12.43 | 22.51 | 4.42 | 4.19 | 7.00 | 13.94 |
| FDS [49] | 132.64 | 109.28 | 311.35 | 851.06 | 7.83 | 7.23 | 12.60 | 22.37 | 4.42 | 4.20 | 6.93 | 13.48 |
| LDS + FDS [49] | 129.35 | 106.52 | 311.49 | 811.82 | 7.78 | 7.20 | 12.61 | 22.19 | 4.37 | 4.12 | 7.39 | 12.61 |
| RANKSIM [13] | 125.30 | 102.68 | 299.10 | 777.48 | 7.50 | 6.93 | 12.09 | 21.68 | 4.19 | 3.97 | 6.65 | 13.28 |
| LDS + FDS + DER [1] | 120.86 | 97.75 | 297.64 | 873.10 | 7.24 | 6.64 | 11.87 | 23.44 | 3.93 | 3.69 | 6.64 | 16.00 |
| VIR (OURS) | 118.94±1.10 | 96.10±0.80 | 295.79±1.20 | 771.47±3.10 | 7.19±0.03 | 6.56±0.03 | 11.81±0.04 | 20.96±0.05 | 3.85±0.04 | 3.63±0.05 | 6.51±0.03 | 12.23±0.03 |
| OURS VS. VANILLA | +19.12 | +12.6 | +70.3 | +193.45 | +0.87 | +0.67 | +3.31 | +5.37 | +0.72 | +0.54 | +4.08 | +8.23 |
| OURS VS. INFER NOISE | +24.68 | +16.16 | +77.40 | +190.50 | +0.92 | +0.80 | +3.42 | +5.33 | +0.83 | +0.70 | +4.14 | +8.08 |
| OURS VS. DER | +14.87 | +11.41 | +37.11 | +144.71 | +0.66 | +0.62 | +1.54 | +3.16 | +0.62 | +0.55 | +1.67 | +2.95 |
| OURS VS. LDS + FDS | +10.41 | +10.42 | +15.7 | +40.35 | +0.59 | +0.64 | +0.8 | +1.23 | +0.52 | +0.49 | +0.88 | +0.38 |
| OURS VS. RANKSIM | +6.36 | +6.58 | +3.31 | +6.01 | +0.31 | +0.37 | +0.28 | +0.72 | +0.34 | +0.34 | +0.14 | +1.05 |

the preprocessing procedures in DIR [49]. Details for each datasets are to the Supplement, and details for label density distributions and levels of imbalance are discussed in DIR [49].

- *AgeDB-DIR*: We use AgeDB-DIR constructed in DIR [49], which contains 12.2K images for training and 2.1K images for validation and testing. The maximum age in this dataset is 101 and the minimum age is 0, and the number of images per bin varies between 1 and 353.
- *IW-DIR*: We use IMDB-WIKI-DIR (IW-DIR) constructed in DIR [49], which contains 191.5K training images and 11.0K validation and testing images. The maximum age is 186 and minimum age is 0; the maximum bin density is 7149, and minimum bin density is 1.
- *STS-B-DIR*: We use STS-B-DIR constructed in DIR [49], which contains 5.2K pairs of training sentences and 1.0K pairs for validation and testing. This dataset is a collection of sentence pairs generated from news headlines, video captions, etc. Each pair is annotated by multiple annotators with a similarity score between 0 and 5.
- *NYUD2-DIR*: We use NYUD2-DIR constructed in DIR [49], which contains 50K images for training and 654 images for testing, and to make the test set balanced 9357 test pixels for each bin are randomly selected. The depth maps have an upper bound of 10 meters, and we set the bin length as 0.1 meter.

**Baselines.** We use ResNet-50 [17] (for AgeDB-DIR, IMDB-WIKI-DIR and NYUD2-DIR) and BiLSTM [19] (for STS-B-DIR) as our backbone networks, and moredetails for baseline are in the supplement. we describe the baselines below.

- *Vanilla*: We use the term **VANILLA** to denote a plain model without adding any approaches.

Table 9: **Accuracy on STS-B-DIR.** For baselines, we directly use the reported performance in their paper and therefore do not have error bars.

| Metrics | MSE ↓ | | | | MAE ↓ | | | | Pearson ↑ | | | | Spearman ↑ | | | |
|---|---|---|---|---|---|---|---|---|---|---|---|---|---|---|---|---|
| Shot | All | Many | Medium | Few | All | Many | Medium | Few | All | Many | Medium | Few | All | Many | Medium | Few |
| VANILLA [49] | 0.974 | 0.851 | 1.520 | 0.984 | 0.794 | 0.740 | 1.043 | 0.771 | 0.742 | 0.720 | 0.627 | 0.752 | 0.744 | 0.688 | 0.505 | 0.750 |
| VAE [25] | 0.968 | 0.833 | 1.511 | 1.102 | 0.782 | 0.721 | 1.040 | 0.767 | 0.751 | 0.724 | 0.621 | 0.749 | 0.752 | 0.674 | 0.501 | 0.743 |
| DEEP ENSEMBLE [27] | 0.972 | 0.846 | 1.496 | 1.032 | 0.791 | 0.723 | 1.096 | 0.792 | 0.746 | 0.723 | 0.619 | 0.750 | 0.741 | 0.689 | 0.501 | 0.746 |
| INFER NOISE [29] | 0.954 | 0.980 | 1.408 | 0.967 | 0.795 | 0.745 | 0.977 | 0.741 | 0.747 | 0.711 | 0.631 | 0.756 | 0.742 | 0.681 | 0.508 | 0.753 |
| SMOTER [40] | 1.046 | 0.924 | 1.542 | 1.154 | 0.834 | 0.782 | 1.052 | 0.861 | 0.726 | 0.693 | 0.653 | 0.706 | 0.726 | 0.656 | 0.556 | 0.691 |
| SMOGN [5] | 0.990 | 0.896 | 1.327 | 1.175 | 0.798 | 0.755 | 0.967 | 0.848 | 0.732 | 0.704 | 0.655 | 0.692 | 0.732 | 0.670 | 0.551 | 0.670 |
| INV [49] | 1.005 | 0.894 | 1.482 | 1.046 | 0.805 | 0.761 | 1.016 | 0.780 | 0.728 | 0.703 | 0.625 | 0.732 | 0.731 | 0.672 | 0.541 | 0.714 |
| DER [1] | 1.001 | 0.912 | 1.368 | 1.055 | 0.812 | 0.772 | 0.989 | 0.809 | 0.732 | 0.711 | 0.646 | 0.742 | 0.731 | 0.672 | 0.519 | 0.739 |
| LDS [49] | 0.914 | 0.819 | 1.319 | 0.955 | 0.773 | 0.729 | 0.970 | 0.772 | 0.756 | 0.734 | 0.638 | 0.762 | 0.761 | 0.704 | 0.556 | 0.743 |
| FDS [49] | 0.927 | 0.851 | 1.225 | 1.012 | 0.771 | 0.740 | 0.914 | 0.756 | 0.750 | 0.724 | 0.667 | 0.742 | 0.752 | 0.692 | 0.552 | 0.748 |
| LDS + FDS [49] | 0.907 | 0.802 | 1.363 | 0.942 | 0.766 | 0.718 | 0.986 | 0.755 | 0.760 | 0.740 | 0.652 | 0.766 | 0.764 | 0.707 | 0.549 | 0.749 |
| RANKSIM [13] | 0.903 | 0.908 | 0.911 | 0.804 | 0.761 | 0.759 | 0.786 | 0.712 | 0.758 | 0.706 | 0.690 | 0.827 | 0.758 | 0.673 | 0.493 | 0.849 |
| LDS + FDS + DER [1] | 1.007 | 0.880 | 1.535 | 1.086 | 0.812 | 0.757 | 1.046 | 0.842 | 0.729 | 0.714 | 0.635 | 0.731 | 0.730 | 0.680 | 0.526 | 0.699 |
| VIR (OURS) | **0.892**±0.002 | **0.795**±0.002 | **0.899**±0.002 | **0.781**±0.003 | **0.740**±0.002 | **0.706**±0.001 | **0.779**±0.002 | **0.708**±0.002 | **0.776**±0.004 | **0.752**±0.003 | **0.696**±0.005 | **0.845**±0.006 | **0.775**±0.003 | **0.716**±0.003 | **0.586**±0.005 | **0.861**±0.007 |
| OURS vs. VANILLA | +0.082 | +0.056 | +0.621 | +0.203 | +0.054 | +0.034 | +0.264 | +0.063 | +0.034 | +0.032 | +0.069 | +0.093 | +0.031 | +0.028 | +0.081 | +0.111 |
| OURS vs. INFER NOISE | +0.062 | +0.185 | +0.509 | +0.186 | +0.055 | +0.039 | +0.198 | +0.033 | +0.029 | +0.041 | +0.065 | +0.089 | +0.033 | +0.035 | +0.078 | +0.108 |
| OURS vs. DER | +0.109 | +0.117 | +0.469 | +0.274 | +0.072 | +0.066 | +0.210 | +0.101 | +0.044 | +0.041 | +0.050 | +0.103 | +0.044 | +0.044 | +0.067 | +0.122 |
| OURS vs. LDS + FDS | +0.015 | +0.007 | +0.464 | +0.161 | +0.026 | +0.012 | +0.207 | +0.047 | +0.016 | +0.012 | +0.044 | +0.079 | +0.011 | +0.009 | +0.037 | +0.112 |
| OURS vs. RANKSIM | +0.011 | +0.113 | +0.012 | +0.023 | +0.021 | +0.053 | +0.007 | +0.004 | +0.018 | +0.046 | +0.006 | +0.018 | +0.017 | +0.043 | +0.093 | +0.012 |

Table 10: Accuracy on NYUD2-DIR.

| Metrics | RMSE ↓ | | | | $\log_{10}$ ↓ | | | | $\delta_1$ ↑ | | | | $\delta_2$ ↑ | | | | $\delta_3$ ↑ | | | |
|---|---|---|---|---|---|---|---|---|---|---|---|---|---|---|---|---|---|---|---|---|
| Shot | All | Many | Medium | Few | All | Many | Medium | Few | All | Many | Medium | Few | All | Many | Medium | Few | All | Many | Medium | Few |
| VANILLA [49] | 1.477 | 0.591 | 0.952 | 2.123 | 0.086 | 0.066 | 0.082 | 0.107 | 0.677 | 0.777 | 0.693 | 0.570 | 0.899 | 0.956 | 0.906 | 0.840 | 0.969 | 0.990 | 0.975 | 0.946 |
| VAE [25] | 1.483 | 0.596 | 0.949 | 2.131 | 0.084 | 0.062 | 0.079 | 0.110 | 0.675 | 0.774 | 0.693 | 0.575 | 0.894 | 0.951 | 0.906 | 0.846 | 0.963 | 0.982 | 0.976 | 0.951 |
| DEEP ENSEMBLE [27] | 1.479 | 0.595 | 0.954 | 2.126 | 0.091 | 0.067 | 0.082 | 0.109 | 0.678 | 0.782 | 0.702 | 0.583 | 0.906 | 0.961 | 0.912 | 0.851 | 0.972 | 0.993 | 0.981 | 0.956 |
| INFER NOISE [29] | 1.480 | 0.594 | 0.959 | 2.125 | 0.088 | 0.069 | 0.089 | 0.111 | 0.672 | 0.768 | 0.688 | 0.566 | 0.894 | 0.949 | 0.902 | 0.834 | 0.963 | 0.983 | 0.970 | 0.941 |
| DER [1] | 1.483 | 0.615 | 0.961 | 2.142 | 0.098 | 0.089 | 0.091 | 0.110 | 0.597 | 0.647 | 0.657 | 0.525 | 0.880 | 0.904 | 0.894 | 0.851 | 0.964 | 0.974 | 0.959 | 0.955 |
| LDS [49] | 1.387 | 0.671 | 0.913 | 1.954 | 0.086 | 0.079 | 0.079 | 0.097 | 0.672 | 0.701 | 0.706 | 0.630 | 0.907 | 0.932 | 0.929 | 0.875 | 0.976 | 0.984 | 0.982 | 0.964 |
| FDS [49] | 1.442 | 0.615 | 0.940 | 2.059 | 0.084 | 0.069 | 0.080 | 0.101 | 0.681 | 0.760 | 0.695 | 0.596 | 0.903 | 0.952 | 0.918 | 0.849 | 0.975 | 0.989 | 0.976 | 0.960 |
| LDS + FDS [49] | 1.338 | 0.670 | 0.851 | 1.880 | 0.080 | 0.074 | 0.070 | 0.090 | 0.705 | 0.730 | 0.764 | 0.655 | 0.916 | 0.939 | 0.941 | 0.884 | 0.979 | 0.984 | 0.983 | 0.971 |
| LDS + FDS + DER [1] | 1.426 | 0.703 | 0.906 | 1.918 | 0.092 | 0.081 | 0.088 | 0.098 | 0.676 | 0.677 | 0.754 | 0.621 | 0.889 | 0.912 | 0.899 | 0.862 | 0.964 | 0.976 | 0.969 | 0.958 |
| VIR (OURS) | **1.305** | **0.589** | **0.831** | **1.749** | **0.075** | **0.060** | **0.064** | **0.082** | **0.722** | **0.781** | **0.793** | **0.688** | **0.929** | **0.966** | **0.961** | **0.910** | **0.985** | **0.993** | **0.989** | **0.979** |
| OURS vs. VANILLA | +0.172 | +0.002 | +0.121 | +0.374 | +0.011 | +0.006 | +0.018 | +0.025 | +0.045 | +0.004 | +0.100 | +0.118 | +0.003 | +0.010 | +0.055 | +0.070 | +0.016 | +0.003 | +0.014 | +0.033 |
| OURS vs. INFER NOISE | +0.175 | +0.005 | +0.128 | +0.376 | +0.013 | +0.009 | +0.025 | +0.029 | +0.050 | +0.013 | +0.105 | +0.122 | +0.035 | +0.017 | +0.059 | +0.076 | +0.022 | +0.010 | +0.019 | +0.038 |
| OURS vs. DER | +0.178 | +0.026 | +0.130 | +0.393 | +0.023 | +0.029 | +0.027 | +0.028 | +0.125 | +0.134 | +0.136 | +0.163 | +0.049 | +0.062 | +0.067 | +0.059 | +0.021 | +0.019 | +0.030 | +0.024 |
| OURS vs. LDS + FDS | +0.033 | +0.081 | +0.020 | +0.131 | +0.005 | +0.014 | +0.006 | +0.008 | +0.017 | +0.051 | +0.029 | +0.033 | +0.013 | +0.027 | +0.020 | +0.026 | +0.006 | +0.009 | +0.006 | +0.008 |

- *Synthetic-Sample-Based Methods*: Various existing imbalanced regression methods are also included as baselines; these include Deep Ensemble [27], Infer Noise [29], SMOTER [40], and SMOGN [5].
- *Cost-Sensitive Reweighting*: As shown in DIR [49], the square-root weighting variant (SQINV) baseline (i.e. $\left( \sum_{b' \in \mathcal{B}} k(y_b, y_{b'}) p(y_{b'}) \right)^{-1/2}$) always outperforms Vanilla. Therefore, for simplicity and fair comparison, *all* our experiments (for both baselines and VIR) use SQINV weighting. To use SQINV in VIR, one simply needs to use the symmetric kernel $k(\cdot, \cdot)$ described in the Method section of the main paper. To use SQINV in DER, we replace the final layer in DIR [49] with the DER layer [1] to produce the predictive distributions.

**Evaluation Process.** Following [28, 49], for a data sample $x_i$ with its label $y_i$ which falls into the target bins $b_i$, we divide the label space into three disjoint subsets: many-shot region $\{b_i \in \mathcal{B} \mid y_i \in b_i \ \& \ |y_i| > 100\}$, medium-shot region $\{b_i \in \mathcal{B} \mid y_i \in b_i \ \& \ 20 \leq |y_i| \leq 100\}$, and few-shot region $\{b_i \in \mathcal{B} \mid y_i \in b_i \ \& \ |y_i| < 20\}$, where $|\cdot|$ denotes the cardinality of the set. We report results on the overall test set and these subsets with the accuracy metrics discussed above.

**Implementation Details.** We use ResNet-50 [17] for all experiments in AgeDB-DIR and IMDB-WIKI-DIR. For all the experiments in STS-B-DIR, we use 300-dimensional GloVe word embeddings (840B Common Crawl version) [31] (following [41]) and a two-layer, 1500-dimensional (per direction) BiLSTM [19] with max pooling to encode the paired sentences into independent vectors $u$ and $v$, and then pass $[u; v; |u - v|; uv]$ to a regressor. We use the Adam optimizer [24] to train all models for 100 epochs, with same learning rate and decay by 0.1 and the 60-th and 90-th epoch, respectively. In order to determine the optimal batch size for training, we try different batch sizes and corroborate the conclusion from [49], i.e., the optimal batch size is 256 when other hyperparameters are fixed. Therefore, we stick to the batch size of 256 for all the experiments in the paper. We also use the same configuration as in DIR [49] for other hyperparameters.

We use PyTorch to implement our method. For fair comparison, we implemented a PyTorch version for the official TensorFlow implementation of DER[1]. To make sure we can obtain the reasonable uncertainty estimations, we restrict the range for $\alpha$ to $[1.5, \infty)$ instead of $[1.0, \infty)$ in DER. Besides, in the activation function *SoftPlus*, we set the hyperparameter *beta* to 0.1. As discussed in the main paper, we implement a layer which produces the parameters $n, \Psi, \Phi$. We assign 2 as the minimum number for $n$, and use the same hyperparameter settings for activation function for DER layer.

Table 11: Uncertainty on NYUD2-DIR.

| Metrics | NLL ↓ | | | | AUSE ↓ | | | |
|---|---|---|---|---|---|---|---|---|
| Shot | All | Many | Medium | Few | All | Many | Medium | Few |
| DEEP ENSEMBLE [27] | 5.054 | 3.640 | 3.856 | 5.335 | 0.782 | 0.658 | 0.604 | 0.583 |
| INFER NOISE [29] | 4.542 | 3.120 | 3.634 | 5.028 | 0.764 | 0.643 | 0.566 | 0.408 |
| DER [1] | 4.169 | 2.913 | 3.011 | 4.777 | 0.713 | 0.623 | 0.535 | 0.382 |
| LDS + FDS + DER [1] | 4.175 | 2.987 | 2.976 | 4.686 | 0.715 | 0.629 | 0.511 | 0.366 |
| VIR (OURS) | **3.866** | **2.815** | **2.727** | **4.113** | **0.690** | **0.603** | **0.493** | **0.335** |
| OURS VS. DER | +0.303 | +0.098 | +0.284 | +0.664 | +0.023 | +0.020 | +0.042 | +0.047 |
| OURS VS. LDS + FDS + DER | +0.309 | +0.172 | +0.249 | +0.573 | +0.025 | +0.026 | +0.018 | +0.031 |

Table 12: Ablation study on $\lambda$ on AgeDB-DIR.

| Metrics | MSE ↓ | | | | MAE ↓ | | | | NLL ↓ | | | |
|---|---|---|---|---|---|---|---|---|---|---|---|---|
| Shot | All | Many | Medium | Few | All | Many | Medium | Few | All | Many | Medium | Few |
| $\lambda = 10.0$ | 104.31 | 91.01 | 116.43 | 196.35 | 7.88 | 7.38 | 8.42 | 11.13 | 3.827 | 3.733 | 4.140 | 4.407 |
| $\lambda = 1.0$ | 104.10 | 87.28 | 128.26 | 196.12 | 7.83 | 7.21 | 8.81 | 10.89 | 3.848 | 3.738 | 4.041 | 4.356 |
| $\lambda = 0.1$ | 86.28 | 76.87 | 101.57 | 132.90 | 7.19 | 6.75 | 7.97 | 9.19 | 3.785 | 3.694 | 3.963 | 4.151 |
| $\lambda = 0.01$ | 86.86 | 76.58 | 99.95 | 147.82 | 7.12 | 6.69 | 7.72 | 9.59 | 3.887 | 3.797 | 4.007 | 4.401 |
| $\lambda = 0.001$ | 87.25 | 74.13 | 104.78 | 162.64 | 7.13 | 6.64 | 7.92 | 9.63 | 3.980 | 3.868 | 4.161 | 4.546 |

To search for a combination hyperparameters of prior distribution $\{\gamma_0, \nu_0, \alpha_0, \beta_0\}$ for NIG, we combine grid search method and random search method [2] to select the best hyperparameters. We first intuitively assign a value and a proper range with some step sizes which correspond to the hyperparameters, then, we apply grid search to search for the best combination for the hyperparameters on prior distributions. After locating a smaller range for each hyperparameters, we use random search to search for better combinations, if it exists. In the end, we find our best hyperparameter combinations for NIG prior distributions.

## C  Complete Results

We include the complete results for all the experiments in AgeDB-DIR, IMDB-WIKI-DIR, STS-B-DIR and NYUD2-DIR in Table 7, Table 8, Table 9 and Table 10. These results demonstrate the superiority of our methods. Note that we did not select to report the baseline for **DIR + DEEP ENS.** since in DER paper [1], it has been showed that DER is better than Deep Ensemble method, therefore we select to report the baseline **DIR+DER** rather than **DIR + DEEP ENS.**.

## D  Discussions

### D.1  Why We Need Bins

Throughout our method, we need to compute the statistics (i.e., the mean and variance) and the "statistics of statistics" of data points (Line 164-165); computing these statistics (e.g., the mean) requires a group of data points. Therefore, we need to partition the continuous label size into $\mathcal{B}$ bins. For example, in the equations from Line 176-177, e.g., $\mu_b^\mu = \frac{1}{N_b} \sum_{i=1}^{N_b} z_i^\mu$, we need to compute the statistics of bin $b$, which contains $N_b$ data points in the bin.

It is also worth noting in the extreme case where (i) each data point has a different label $y$ and (ii) we use a very small bin size, each bin will then contain exactly only one data point.

### D.2  Equal-Interval Bins versus Equal-Size Bins

Note that since our smoothing kernel function is based on labels (i.e., $k(y, y')$), it is more reasonable to use **equal-interval** bins rather than **equal-size** bins.

Table 13: **Comparison for different numbers of Bins.** "Med." is short for "Medium".

| Metrics | Bins | MSE ↓ | | | | MAE ↓ | | | | GM ↓ | | | |
|---|---|---|---|---|---|---|---|---|---|---|---|---|---|
| Shot | # | All | Many | Med. | Few | All | Many | Med. | Few | All | Many | Med. | Few |
| RANKSIM | 100 | 83.51 | 71.99 | 99.14 | 149.05 | 7.02 | 6.49 | 7.84 | 9.68 | 4.53 | 4.13 | 5.37 | 6.89 |
| VIR (OURS) | 100 | **81.76** | **70.61** | **91.47** | **142.36** | **6.99** | **6.39** | **7.47** | **9.51** | **4.41** | **4.07** | **5.05** | **6.23** |
| RANKSIM | 33 | 109.45 | 91.78 | 128.10 | 187.13 | 7.46 | 6.94 | 8.42 | 10.66 | 5.13 | 4.70 | 5.23 | 8.21 |
| VIR (OURS) | 33 | **84.77** | **77.29** | **95.66** | **125.33** | **7.01** | **6.70** | **7.45** | **8.74** | **4.36** | **4.20** | **4.73** | **4.94** |
| RANKSIM | 20 | 98.71 | 84.38 | 107.89 | 171.04 | 7.32 | 6.78 | 8.35 | 10.57 | 5.33 | 4.51 | 5.69 | 7.92 |
| VIR (OURS) | 20 | **84.05** | **72.12** | **100.49** | **151.25** | **7.06** | **6.50** | **7.90** | **10.06** | **4.49** | **4.05** | **5.34** | **7.28** |

- For example, if we use the equal-interval bins $[0, 1), [1, 2), ...$, VIR will naturally compute $k(y, y')$ for $y = 1, 2, 3, 4, 5, ...$ and $y' = 1, 2, 3, 4, 5, ...$.
- In contrast, if we use equal-size bins, VIR may end up with **large intervals** and may lead to inaccurate kernel values for $k(y, y')$. To see this, consider a case where equal-size bins are $[0, 1), [1, 2), [2, 3.1), [3.1, 8.9), ...$; the kernel value $k(y, y')$ between bins $[2, 3.1)$ and $[3.1, 8.9)$ is $k(2, 3.1)$, which is very inaccurate since 3.1 is very far away from the mean of the bin $[3.1, 8.9)$ (i.e., 6). Using small and equal-interval bins can naturally address such issues.

### D.3 The Number of Bins

Our preliminary results indicate that the performance of our VIR remains consistent regardless of the number of bins, as shown in the Sec. E.3 of the Supplement. Thus in our paper, we chose to use the same number of bins as the imbalanced regression literature [13, 49] for fair comparison with prior work. For example, in the AgeDB dataset where the regression labels are people's "age" in the range of 0 99, we use 100 bins, with each year as one bin.

### D.4 Reweighting Methods and Stronger Augmentations

Our method focus on reweighting methods, and using augmentation (e.g., the SimCLR pipeline [10]) is an orthogonal direction to our work. However, we expect that data augmentation could further improve our VIR's performance. This is because one could perform data augmentation only on minority data to improve accuracy in the minority group, but this is sub-optimal; the reason is that one could potentially further perform data augmentation on majority data to improve accuracy in the majority group without sacrificing too much accuracy in the minority group. However, performing data augmentation on both minority and majority groups does not transform an imbalanced dataset to an balanced dataset. This is why our VIR is still necessary; VIR could be used on top of any data augmentation techniques to address the imbalance issue and further improve accuracy.

### D.5 Discussion on I.I.D. and N.I.D. Assumptions

**Generalization Error, Bias, and Variance.** We could analyze the generalization error of our VIR by bounding the generalization with the sum of three terms: (a) the bias of our estimator, (2) the variance of our estimator, (3) model complexity. Essentially VIR uses the N.I.D. assumption increases our estimator's bias, but significantly reduces its variance in the imbalanced setting. Since the model complexity is kept the same (using the same backbone neural network) as the baselines, N.I.D. will lead to a lower generalization error.

**Variance of Estimators in Imbalanced Settings.** In the imbalanced setting, one typically use inverse weighting (i.e., the IPS estimator in Definition A.4) to produced an unbiased estimator (i.e., making the first term of the aforementioned bound zero). However, for data with extremely low density, its inverse would be extremely large, therefore leading to a very large variance for the estimator. Our VIR replaces I.I.D. with N.I.D. to "smooth out" such singularity, and therefore significantly lowers the variance of the estimator (i.e., making the second term of the aforementioned bound smaller), and ultimately lowers the generalization error.

## D.6 Why We Need Statistics of Statistics for Smoothing

Compared with DIR [49], which only considers the **statistics** for *deterministic representations*, our VIR considers the **statistics of statistics** for *probabilistic representations*, this is because the requirement to perform feature smoothing to get the representation $z_i$ necessitates the computation of mean and variance of $z_i$'s neighboring data (i.e., data with neighboring labels). Here $z_i$ contains the **statistics** of neighboring data. In contrast, our VIR also needs to generate uncertainty estimation, which requires a stochastic representation for $z_i$, e.g., the mean and variance of $z_i$ (note that $z_i$ itself is already a form of statistics). This motivates the hierarchical structure of the **statistics of statistics**. Here the variance measures the uncertainty of the representation.

## D.7 The Choice of Kernel Function

The DIR paper shows that a simple Gaussian kernel with inverse square-root weighting (i.e., SQINV) achieves the best performance. Therefore, we use exactly the same parameter configuration as the DIR paper [49]. Specifically, we set $\sigma = 2$; for label $y_b$ in bin $b$, we define neighboring labels as labels $y_{b'}$ such that $|y_{b'} - y_b| \leq 2$, i.e., $B$ contains 5 bins. For example, if $y_b = 23$, its neighboring labels are 21, 22, 23, 24, and 25.

Besides, our preliminary results also show that the performance is not very sensitive to the choice of kernels, as long as the kernel $k(a, b)$ reflects the distance between $a$ and $b$, i.e., larger distance between $a$ and $b$ leads to smaller $k(a, b)$.

## D.8 Why VIR Solves the Imbalanced Regression Problem

Our training objective function (Eqn.5 in the main paper) is the **negative log likelihood** for the Normal Inverse Gaussian (NIG) distribution, and each posterior parameter $(\nu_i^*, \gamma_i^*, \alpha_i^*)$ of the NIG distribution is reweighted by importance weights, thereby assigning higher weights to minority data during training and allowing minority data points to benefit more from their neighboring information.

Take $\nu_i^*$ as an example. Assume a minority data point $(x_i, y_i)$ that belongs to bin $b$, i.e., its label $y_i = y_b$. Note that there is **a loss term** $(y_i - \gamma_i^*)^2 \nu_i^*$ in **Eqn.5**, where $\gamma_i^*$ is the model prediction, $y_i$ is the label, and $\nu_i^*$ is the *importance weight* for this data point.

Here $\nu_i^* = \nu_0 + (\sum_{b' \in \mathcal{B}} k(y_b, y_{b'}) p(y_{b'}))^{-1/2} \cdot n_i$ where $n_i$ represents the pseudo-count for the NIG distribution. Since $(x_i, y_i)$ is a minority data point, data from its neighboring bins has smaller frequency $p(y_{b'})$ and therefore smaller $\sum_{b' \in \mathcal{B}} k(y_b, y_{b'}) p(y_{b'})$, leading to **a larger *importance weight* $\nu_i^*$ for this minority data point in Eqn.5**.

This allows VIR to naturally put more focus on the minority data, thereby alleviating the imbalance problem.

## D.9 Difference from DIR, VAE and DER

From a technical perspective, VIR is substantially different from any combinations of DIR [49], VAE [25], and DER [1]. Specifically,

- VIR is a deep generative model to define how imbalanced data are generated, which is learned by a principled variational inference algorithm. In contrast, DIR is a simply discriminative model (without any principled generative model formulation) that directly predict the labels from input. It is more prone to overfitting.
- DIR uses deterministic representations, with one vector as the final representation for each data point. In contrast, our VIR uses probabilistic representations, with one vector as the mean of the representation and another vector as the variance of the representation. Such dual representation is more robust to noise and therefore leads to better prediction performance.
- DIR is a deterministic model, while our VIR is a Bayesian model. Essentially VIR is equivalent to sampling infinitely many predictions for each input data point and averaging these predictions. Therefore intuitively it makes sense that VIR could lead to better prediction performance.

- Different from VAE and DIR, VIR introduces a reweighting mechanism naturally through the pseudo-count formulation in the NIG distribution (discussed in the paragraphs **Intuition of Pseudo-Counts for VIR** and **From Pseudo-Counts to Balanced Predictive Distribution** in the paper). Note that such a reweighting mechanism is more natural and powerful than DIR since it is rooted in the probabilistic formulation.
- Unlike for the standard VAE, the optimal prior (that maximizes ELBO) is known to be the aggregated posterior, our optimal prior is a *neighbor-weighted* version of aggregated posterior: for standard VAE, different data points contribute **independently** to the aggregated posterior; in contrast, for our VIR, the importance of each data point with respect to the aggregated posterior is **affected** by data points with neighboring labels.
- It is also worth noting that DIR cannot produce uncertainty estimation since it is a deterministic model. In contrast, Our VIR formulates a probabilistic deep generative model for imbalanced data, and therefore can naturally produce both more accurate predictions compared to DIR and better uncertainty estimation compared to DER.

### D.10   Bayesian Neural Networks (BNNs)

We do not consider other BNNs in this work because:

- Weights in Bayesian Neural Networks (BNNs) are extremely high-dimensional; therefore BNNs have several limitations, including the intractability of directly inferring the posterior distribution of the weights given data, the requirement and computational expense of sampling during inference, and the question of how to choose a weight prior [1]. In contrast, evidential regression does not have these challenges.
- In our preliminary experiments, we found that typical BNN methods suffer from computational inefficiency and would require at least two to three times more computational time and memory usage. In contrast, evidential regression does not involve such computation and memory overhead; its overhead only involves the last (few) layers, and is therefore minimal.
- Additionally, as demonstrated in [27], Deep Ensemble typically performs as well as or even better than BNNs. Our method outperforms Deep Ensemble, therefore suggesting its superiority over typical BNN methods.

## E   Additional Experiment Results

### E.1   Ablation Study on VIR

In this section, we include ablation studies to verify that our VIR can outperform its counterparts in DIR (i.e., smoothing on the latent space) and DER (i.e., NIG distribution layers).

**Ablation Study on $q(\mathbf{z}_i|\{\mathbf{x}_i\}_{i=1}^N)$.** To verify the effectiveness of VIR's encoder $q(\mathbf{z}_i|\{\mathbf{x}_i\}_{i=1}^N)$, we replace VIR's predictor $p(y_i|\mathbf{z}_i)$ with a linear layer (as in DIR). Table 14 shows that compared to its counterpart, FDS [49], our encoder-only VIR leads to a considerable improvements even without generating the NIG distribution. Both verify the effectiveness of our VIR's $q(\mathbf{z}_i|\{\mathbf{x}_i\}_{i=1}^N)$.

Table 14: Ablation study for Accuracy.

| Metrics | MSE ↓ | | | | MAE ↓ | | | |
|---|---|---|---|---|---|---|---|---|
| Shot | All | Many | Med. | Few | All | Many | Med. | Few |
| FDS [49] | 109.78 | 93.99 | 124.96 | 216.97 | 8.12 | 7.52 | **8.68** | 12.25 |
| **ENCODER-ONLY VIR (OURS)** | 95.99 | 81.89 | 121.78 | 157.92 | 7.57 | 6.97 | 8.72 | 10.03 |
| DER [1] | 106.81 | 91.32 | 122.45 | 209.76 | 8.11 | 7.36 | 9.03 | 12.69 |
| **PREDICTOR-ONLY VIR (OURS)** | 88.96 | 74.79 | 95.85 | 203.76 | 7.28 | 6.68 | 7.76 | 11.63 |
| LDS+FDS | 99.46 | 84.10 | 112.20 | 209.27 | 7.55 | 7.01 | 8.24 | 10.79 |
| LDS + **PREDICTOR-ONLY VIR (OURS)** | 87.48 | 73.72 | 107.64 | 161.69 | 7.17 | 6.63 | 8.06 | 9.80 |
| LDS + **ENCODER-ONLY VIR (OURS)** | 96.46 | 86.72 | 102.56 | 171.52 | 7.51 | 7.08 | 7.93 | 10.45 |
| **VIR (OURS)** | **81.76** | **70.61** | **91.47** | **142.36** | **6.99** | **6.39** | **7.47** | **9.51** |

**Ablation Study on $p(y_i|\mathbf{z}_i)$.** To verify the effectiveness of VIR's predictor $p(y_i|\mathbf{z}_i)$, we replace VIR's encoder $q(\mathbf{z}_i|\{\mathbf{x}_i\}_{i=1}^N)$ with a simple deterministic encoder as in DER [1]. Table 14 and Table 15 show that compared to DER, the counterpart of VIR's predictor, our VIR's predictor still outperforms than DER, demonstrating its effectiveness. Both verifies our claim that directly reweighting DER breaks NIG and leads to poor performance.

Table 15: Ablation study for Uncertainty.

| Metrics | NLL ↓ | | | | AUSE ↓ | | | |
|---|---|---|---|---|---|---|---|---|
| Shot | All | Many | Med. | Few | All | Many | Med. | Few |
| DER [1] | 3.936 | 3.768 | 3.865 | 4.421 | 0.590 | 0.449 | 0.468 | 0.500 |
| **PREDICTOR-ONLY VIR (OURS)** | **3.887** | **3.755** | **3.854** | **4.394** | **0.443** | **0.387** | **0.390** | **0.407** |
| LDS + **PREDICTOR-ONLY VIR (OURS)** | 3.722 | 3.604 | 3.821 | 4.209 | 0.441 | 0.457 | 0.334 | 0.426 |
| **VIR (OURS)** | **3.703** | **3.598** | **3.805** | **4.196** | **0.434** | 0.456 | **0.324** | **0.414** |

## E.2   Ablation Study on $\lambda$.

In this section, we include ablation studies on the $\lambda$ in our objective function. For $\lambda \in \{10.0, 1.0, 0.1, 0.01, 0.001\}$, we run our VIR model on the AgeDB dataset. Table 12 shows the results. We can observe that our model achieves the best performance when $\lambda = 0.1$.

## E.3   Ablation Study on Number of Bins

In this section, we include ablation studies on the number of bins in our settings. For the cases with $100/1 = 100$, $100/3 \approx 33$, and $100/5 = 20$ bins, we run our VIR model on the AgeDB dataset. Table 13 shows the results. We can observe that our VIR remains consistent regardless of the number of bins.

Table 16: Calibration Uncertainty on AgeDB-DIR.

| Metrics | NLL $\downarrow$ | | | | AUSE $\downarrow$ | | | |
|---|---|---|---|---|---|---|---|---|
| Shot | All | Many | Medium | Few | All | Many | Medium | Few |
| DEEP ENS. [27] | 5.311 | 4.031 | 6.726 | 8.523 | 0.541 | 0.626 | 0.466 | 0.483 |
| [CALIBRATED] DEEP ENS. [27] | 5.015 | 3.978 | 6.402 | 8.393 | 0.506 | 0.591 | 0.386 | 0.402 |
| INFER NOISE [29] | 4.616 | 4.413 | 4.866 | 5.842 | 0.465 | 0.458 | 0.457 | 0.496 |
| [CALIBRATED] INFER NOISE [29] | 4.470 | 4.183 | 4.756 | 5.622 | 0.404 | 0.426 | 0.410 | 0.415 |
| DER [1] | 3.918 | 3.741 | 3.919 | 4.432 | 0.523 | 0.464 | 0.449 | 0.486 |
| [CALIBRATED] DER [1] | 3.827 | 3.674 | 3.835 | 4.297 | 0.479 | 0.401 | 0.399 | 0.396 |
| LDS + FDS + DER [1] | 3.787 | 3.689 | 3.912 | 4.234 | 0.451 | 0.460 | 0.399 | 0.565 |
| [CALIBRATED] LDS + FDS + DER [1] | 3.708 | 3.636 | 3.807 | 4.032 | 0.417 | 0.364 | 0.269 | 0.452 |
| VIR (OURS) | 3.703 | 3.598 | 3.805 | 4.196 | 0.434 | 0.456 | 0.324 | 0.414 |
| **[CALIBRATED] VIR (OURS)** | **3.577** | **3.493** | **3.595** | **3.866** | **0.359** | **0.232** | **0.276** | **0.266** |

## E.4   Calibrated Uncertainty

In a bid to enhance the evaluation of our model's uncertainty, we used our validation set to apply calibration techniques (specifically, variants of temperature scaling [14]) on different methods. We focused on each of the output distribution parameters $\nu, \alpha, \beta$ as discussed in our main paper, introducing individual scalar weights for each parameter to calibrate uncertainty estimation. Upon deriving each weight $\mathbf{w}_\nu, \mathbf{w}_\alpha, \mathbf{w}_\beta$ from the validation set, we incorporated them into the test dataset to ascertain the final performance. The data outlined in Table 16 indicates that following re-calibration, the uncertainty was further optimized. Notwithstanding such uncertainty calibration, our model persists in demonstrating superior performance compared to other benchmark methods.

## E.5   Error bars on Accuracy

In order to further underscore the superiority of our methodology, we also included error bars for our proposed method (i.e., VIR) which are generated from five independent runs. Note that we did not report error bars for those second/third best baselines (e.g., RankSim, LDS+FDS) since we directly use the reported performance from their papers, and in NYUD2-DIR, all the error bars on VIR are approximately 0.001. Due to the *width* constraints of the paper, these are not included in Table 10. Results in Table 7, Table 8, and Table 9 demonstrate the effectiveness of our approach.

