# OpenReview forum: "Variational Imbalanced Regression: Fair Uncertainty Quantification via Probabilistic Smoothing"
_NeurIPS.cc/2023/Conference — NeurIPS 2023 poster_

### Official Review · Reviewer_C6MR · 2023-07-01

**Soundness:** 3 good
**Presentation:** 3 good
**Contribution:** 3 good
**Rating:** 7
**Confidence:** 4

**Summary:**

This paper simultaneously addresses the label imbalance problem and uncertainty qualification capability in regression.
The authors propose to enhance the reweighting technique dealing with the imbalance problem in (Yang et al., 2021) to be applicable to VAE and combine the method with the output distribution and the corresponding loss in (Amini et al., 2020), which provides uncertainty qualification capability.
Experimental results on several real-world datasets demonstrate that the proposed method performs better than state-of-the-art imbalanced regression methods in terms of both accuracy and uncertainty estimation.

**Strengths:**

### Originality:
- The authors simultaneously address the label imbalance problem and uncertainty estimation capability in regression, which is a novel problem setting.

### Quality:
- The combination of the reweighting technique in (Yang et al., 2021) and the output distribution and the corresponding loss in (Amini et al., 2020) is non-trivial and works well.

- The authors showed the superiority of the proposed method experimentally in terms of both accuracy and uncertainty estimation, where they used multiple public datasets.

### Clarity:
- The presentation is clear.

**Weaknesses:**

- Good combination of the SOTA, but the originality can be limited because of that.

**Questions:**

NA

**Limitations:**

They discussed that the exact computation of variance of the variances is challenging in Section 5.3.

---

> ### Author Rebuttal · Authors · 2023-08-07
>
> Thank you for your constructive and encouraging comments as well as the insightful questions. We are glad that you find the problem we address ``"novel"``, our method ``"non-trivial"``/``"works well"``, our presentation ``"clear"``, and that experiments show our method's ``"superiority"`` ``"in terms of both accuracy and uncertainty estimation"``. Below we address your question.
>
> **Q1: "Good combination of the SOTA, but the originality can be limited because of that."**
>
> Thank you for acknowledging our contribution and our SOTA performance. We should have better highlighted our originality in the main paper. Specifically, our VIR not only combines existing SOTA methods (e.g., DIR), as you mentioned, but also presents substantial divergence from them (see a more in-depth discussion of these differences in Section 1.2 of the Supplementary Material):
>
> **(1)** VIR is a deep generative model to define how imbalanced data are generated, which is learned by a principled variational inference algorithm. In contrast, DIR is a simply discriminative model (without any principled generative model formulation) that directly predicts the labels from input. It is more prone to overfitting.
>
> **(2)** DIR uses deterministic representations, with one vector as the final representation for each data point. In contrast, our VIR uses probabilistic representations, with one vector as the mean of the representation and another vector as the variance of the representation. Such dual representation is more robust to noise and therefore leads to better prediction performance.
>
> **(3)** DIR is a deterministic model, while our VIR is a Bayesian model. Essentially VIR is equivalent to sampling infinitely many predictions for each input data point and averaging these predictions. Therefore intuitively it makes sense that VIR could lead to better prediction performance.
>
> **(4)** Different from VAE and DIR, VIR introduces a reweighting mechanism naturally through the **pseudo-count formulation** in the NIG distribution (discussed in the paragraphs Intuition of Pseudo-Counts for VIR and From Pseudo-Counts to Balanced Predictive Distribution of the paper). Note that such a reweighting mechanism is more natural and powerful than DIR since it is rooted in the probabilistic formulation.
>
> Besides methodological originality, our other contributions include:
>
> **(a)** We identify the problem of probabilistic deep imbalanced regression as well as two desiderata, balanced accuracy and uncertainty estimation, for the problem.
>
> **(b)** As a byproduct, we also provide strong baselines for benchmarking high-quality uncertainty estimation and promising prediction performance on imbalanced datasets.
>
> We will include the discussion above in the main paper of our revision to better highlight our originality as suggested.

---

### Official Review · Reviewer_9XGJ · 2023-07-04

**Soundness:** 4 excellent
**Presentation:** 3 good
**Contribution:** 4 excellent
**Rating:** 8
**Confidence:** 4

**Summary:**

The authors propose a variational regression model for imbalanced data, which (1) borrows data with similar regression labels for variational distribution (neighboring and identically distributed: N.I.D.) and (2) utilize the conjugate distributions to impose probabilistic reweighting on the imbalanced data to give better uncertainty estimation. Experiments show the proposed model achieves the SOTA on several datasets.

**Strengths:**

- Imbalanced regression is a research area that is yet to be explored, although its real-world application is important.

- In addition, the proposed model can provide uncertainty of the prediction, which is important for real-world applications.

- The uncertainty estimation on imbalanced datasets is also an interesting field but is yet to be explored.

- Overall, the present topic is very relevant in the community. I encourage the authors to develop and explore this direction more, in view of the nice performance of the model.

- The performance of the proposed model is excellent. Both of the accuracy and uncertainty estimation outperform the SOTA models.

- The present paper is easy to follow and well-written. I enjoyed reading it. I found some nice "road signs" for the readers to follow the logic and story.

**Weaknesses:**

- The code for reproduction is not available. I would like to see and use your code. I could not find any other major weakness.

- See Questions below.

**Questions:**

Here are my questions and comments.

- [Question] Just out of curiosity. What is the optimal prior for VIR? For the standard VAE, the optimal prior (that maximizes ELBO) is known to be the aggregated posterior. This might lead to an interesting future work.

- [Comment (minor)] "NIG" on L.137 should be spelled out here, rather than on L. 213, or define NIG on L. 40.

- [Comment (super-minor)] (1), (2), ... in the main text might sometimes be confused with the number of equations. Possible alternatives are: (i), (ii),..., (I), (II), ..., (a), (b), ..., (A), (B), ..., 1), 2), ..., etc.

- [Question] Where does the idea of the hierarchical structure of the statistics (L. 166--195) (statistics of statistics for smoothing) come from? What motivated this idea? This is very interesting.

- [Comment (minor)] L. 191: cross -> across?

- [Comment (minor)] L. 212: correspond -> corresponding?

- [Question (major)] At first glance, I could not fully understand why Eqn. 5 alleviate the imbalance problem. Could you clarify the underlying mechanism more?

- [Question (major)] Is the performance improvement (accuracy and uncertainty) "significant"? If so, in what sense? (Sorry for the vague question, but this is an important point when evaluating model's performance and paper's quality because a marginal improvement is sometimes meaningless.)

**Limitations:**

Limitations are included in Section 5.3.

---

> ### Author Rebuttal · Authors · 2023-08-07
>
> Thank you for your constructive and encouraging comments as well as the insightful questions. We are glad that you find the problem we address ``"important"``/``"interesting"``, our performance ``"excellent"``, and our paper ``"easy to follow"``/``"well-written"``. Below we address your questions one by one.
>
> **Q1: "The code for reproduction is not available. I would like to see and use your code. I could not find any other major weakness."**
>
> Thank you very much for your interest. We have finished cleaning up the source code and will release it after the paper is accepted to facilitate further research in the community.
>
> **Q2: "What is the optimal prior for VIR? For the standard VAE, the optimal prior (that maximizes ELBO) is known to be the aggregated posterior. This might lead to an interesting future work"**
>
> Thanks for the insightful question. Our optimal prior is a *neighbor-weighted* version of aggregated posterior: for stanard VAE, different data points contirbute **independetly** to the aggregated posterior; in contrast, for our VIR, the importance of each data point with repspect to the aggregated posterior is **affected by** data points with neighboring labels. We will include the discussion above in the revision to provide more insight into the difference between VAE and VIR.
>
> **Q3: ""NIG" on L.137 should be spelled out here, rather than on L. 213, or define NIG on L. 40."**
>
> Thank you for your suggestion. We will fix this in the revision.
>
> **Q4: "... in the main text might sometimes be confused with the number of equations..."**
>
> Thank you for your suggestion. We will revise our paper accordingly in the revision.
>
> **Q5: "Where does the idea of the hierarchical structure of the statistics (L. 166--195) (statistics of statistics for smoothing) come from? What motivated this idea? This is very interesting."**
>
> This is a good question and thank you for your interest.
>
> **(1) Statistics.** The requirement to perform feature smoothing to get the representation $z_i$ necessitates the computation of mean and variance of $z_i$'s neiboring data (i.e., data with neighboring labels). Here $z_i$ contains the **statistics** of neighboring data.
>
> **(2) Statistics of Statistics.** Furthermore, uncertainty estimation requires a stochastic representation for $z_i$, e.g., the mean and variance of $z_i$ (note that $z_i$ itself is already a form of statistics). This motivates the hierarchical structure of the statistics, i.e., **statistics of statistics**. Here the variance measures the uncertainty of the representation. We appreciate your suggestion and will incorporate this discussion into the revised version of our paper.
>
> **Q6: "L. 191: cross -> across? L. 212: correspond -> corresponding?"**
>
> We are sorry for the typos, and will fix them in the revision.
>
> **Q7: "At first glance, I could not fully understand why Eqn. 5 alleviate the imbalance problem. Could you clarify the underlying mechanism more?"**
>
> We are sorry for the confusion. Please refer to **Q2 in the Global Response**.
>
> **Q8: "Is the performance improvement (accuracy and uncertainty) "significant"? If so, in what sense? (Sorry for the vague question, but this is an important point when evaluating model's performance and paper's quality because a marginal improvement is sometimes meaningless.)"**
>
> Thank you for mentioning this. According to your suggestion, we ran the corresponding hypothesis tests, and the p values are in the range of $(9.08 \times 10^{-7}, 3.38 \times 10^{-4})$, much lower than the threshold of $0.05$ and therefore verifying the significance of VIR's performance improvement. We will include these results in the revision to strengthen the paper as suggested.

---

> > ### Comment · Reviewer_9XGJ · 2023-08-14
> > **Reply by Reviewer 9XGJ**
> >
> > Thank you for the reply. I carefully read all the reviews and responses.
> > - All of my questions and concerns are addressed.
> > - The global response and the discussion with Reviewer cFEB are interesting and will make the paper more convincing.
> > - I think the fact that no other BNN models are not included in the paper is not a major probolem because the proof-of-concept is already done in the main text and the result is convincing.
> > - As for the number of bins, it is a common problem when we use the quantization- (= binning-) based approach to regression problems; it is a common hyperparameter.
> > - The global response shows that the proposed model is relatively robust to the number of bins.
> >
> > Overall, I strongly support the acceptance of the present paper and keep the score as is:
> > - The topic is relevant and important for real-world applications.
> > - The topic is interesting because this is an interdisciplinary research between imbalanced regression and Bayesina inference.
> > - I could not find major technical flaws.
> > - Author's responses are insightful and make sense, which will make the paper better after revision.
> > - The code will be published.
> > - The experimental results are convincing.
> >
> > Excellent work!

---

> > > ### Author Response · Authors · 2023-08-14
> > > **Thank You**
> > >
> > > Thank you for your encouraging and detailed further response! We are glad that you have a thorough understanding of our contributions, find them novel, and acknowledge our rebuttal address all the questions/concerns (including the points on "no other BNNs", "number of bins" and the discussion with Review cFEB). We would be immensely grateful if you could consider raising the confidence score to reflect the current assessment.
> > >
> > > Thank you again!
> > >
> > > Best regards,
> > >
> > > Authors of VIR

---

> > > > ### Comment · Reviewer_9XGJ · 2023-08-21
> > > >
> > > > In light of the detailed and convincing response, I raised the confidence score (3 -> 4).

---

### Official Review · Reviewer_ivhk · 2023-07-04

**Soundness:** 3 good
**Presentation:** 2 fair
**Contribution:** 3 good
**Rating:** 7
**Confidence:** 3

**Summary:**

This paper proposes a variational imbalanced regression model by taking the Neighboring and Identically Distributed (N.I.D.) assumption to solve both imbalanced regression and uncertainty estimation problems. Experiments on four imbalanced datasets demonstrate the effectiveness of the proposed method.

**Strengths:**

1) Compared with imbalanced classification,  imbalanced regression is underexplored and also an important topic.
2) The Neighboring and Identically Distributed (N.I.D.) assumption seems more reasonable than the Independent and Identically Distributed (I.I.D.) assumption.
3) The proposed method improves not only the performance of few-shot region, but also the performance of many-shot region.

**Weaknesses:**

1) The description of some parts of the paper is not clear, e.g., how to define the neighboring labels?  What is the detailed formulation of the importance weights in line 237?
2) Some statements in the paper are somewhat subjective and lack support, e.g., in line 42-43, the authors claim that "This allows the negative log likelihood to naturally put more focus on the minority data", but I do not find any "naturally" thing, the important weights are mainly determined by the kernel functions, which needs manually selection; in line 273, the authors claim that "Such dual representation is more robust to noise", but they neither list any references nor conduct any experiments to support it.

**Questions:**

1) What are the new challenges of imbalanced regression compared with imbalanced classification?
2) Why do you partition the label space into B equal-interval bins? Does the choice of B affect the performance of VIR?
3) The choice of the kernel functions and experiments for the corresponding parameters selection should be given since it determines the importance weights.
4) In section 4.1, the authors claim that N.I.D. will lead to a lower generalization error just by simple text description, without any rigorous computation or derivation, this is not convincing for me.

**Limitations:**

Yes, the authors have adequately addressed the limitations.

---

> ### Author Rebuttal · Authors · 2023-08-07
>
> Thank you for your constructive and encouraging comments with insightful questions. We are glad that you find the problem ``"important"`` and our N.I.D. assumption ``"reasonable"``, and acknowledge that our method improves performance. Below we address your questions in detail one by one.
>
> **Q1: "... how to define the neighboring labels? ... detailed formulation of the importance weights in line 237?"**
>
> We are sorry for the confusion and the typo.
>
> In Line 237~241, it should be $(\sum_{b'\in B}k(y_b,y_{b'})p(y_{b'}))^{-\frac{1}{2}}$ rather than $(\sum_{b'\in B}k(y_b,y_{b'}))^{-\frac{1}{2}}$. Here $B$ is the set of bin $b$'s neighboring bins, $k(y_b,y_{b'})$ is a kernel measuring the distance between labels $y_b$ and $y_{b'}$, and $p(y_{b'})$ is the frequency (density) of label $y_{b'}$ in the dataset. We use a **Gaussian kernel** $k(a, b)=\exp({-\frac{(a - b)^{2}}{2\sigma^2}})$.
>
> For fair comparison, we use exactly the same parameter configuration as the DIR paper [1]. Specifically, we set $\sigma=2$; for label $y_b$ in bin $b$, we define neighboring labels as labels $y_{b'}$ such that $|y_{b'}-y_b|\leq 2$, i.e., $B$ contains $5$ bins. For example, if $y_b=23$, its neighboring labels are $21$, $22$, $23$, $24$, and $25$.
>
> With the definition above, we can see that data in the *minority* neighborhood (smaller $p(y_{b'})$) has *larger* importance weights $(\sum_{b'\in B}k(y_b,y_{b'})p(y_{b'}))^{-\frac{1}{2}}$ in the training objective function.
>
> We will include the details above in the revision as suggested.
>
> **Q2.1: "... allows the negative log likelihood to naturally put more focus on the minority data..."**
>
> We are sorry for the confusion. Please refer to **Q3 in the Global Response**.
>
> **Q2.2: "...needs manually selection;..." "The choice of the kernel functions...should be given..."**
>
> This is a good question. Please refer to **Q1 above** and **Q4 in the Global Response** .
>
> **Q2.3: "..."Such dual representation is more robust to noise"...references...experiments..."**
>
> We are sorry for the confusion. Please refer to **Q2 in the Global Response**.
>
> **Q3: "...new challenges of imbalanced regression compared with imbalanced classification?"**
>
> This is a good question.
>
> As shown in **Figure 1 of the 1-page PDF**, we used two datasets, CIFAR-100, a 100-class classification dataset [2] and IMDB-WIKI [3], an age estimation dataset with labels in the range 0~99, to compare imbalanced data challenges in classification vs. regression. We adjusted their label ranges for consistency and simulated data imbalance, ensuring identical label density distribution, as seen in Fig. 1(a) in our global-response PDF. A ResNet-50 model trained on these datasets highlighted differences in test error distributions.
>
> Results from CIFAR-100 showed a negative correlation between test error and label density ($-0.76$), which is expected since classes with more samples often have lower errors. However, IMDB-WIKI, despite having the same label density as CIFAR-100, had a more uniform error distribution that **did not align as closely with label density ($−0.47$)**, as shown in Fig. 1(b).
>
> *This distinction highlights unique challenges in imbalanced regression*. Traditional imbalanced learning methods, which address the imbalance in the **empirical** label density, work for classification but might falter for regression with continuous labels. The challenge grows in *probabilistic* imbalanced regression, where both accuracy and *uncertainty estimation* matter. Addressing these challenges are therefore the focus of our paper.
>
> **Q4: "Why...partition...into B equal-interval bins? Does the choice of B affect the performance of VIR?"**
>
> This is a good question. Please refer to **Q1 in the Global Response**.
>
> **Q5: "In section 4.1, the authors claim that N.I.D. will lead to a lower generalization error just by simple text description, without any rigorous computation or derivation, this is not convincing for me."**
>
> We are sorry for the confusion. We meant to use Section 4.1 to discuss *intuition* of our methods. We have prepared a rigorous proof and will include it in the Supplementary Material of our revision. Below we discuss the main idea of the proof.
>
> In general, the generalization error is bounded with probability $1-\eta$ by: test error $\leq$ training error + bias + variance, where
>
> bias $=\frac{\Delta}{N} \sum_{(x,y)} |1 - \frac{P_{y}}{\hat{P}_{y}}|$,
>
> variance $=\frac{\Delta}{N} \sqrt{\frac{\log ( 2 |{H}| / \eta)}{2}} \sqrt{\sum_{(x,y)}(\hat{P}_{y})^{-2}}$.
>
> Here
>
> (1) $\hat{P}_{y}$ is the smoothed label distribution used in our VIR's objective function,
>
> (2) $P_{y}$ is the label distribution,
>
> (3) $N$ is the number of data points,
>
> (4) $\Delta=y_{max}-y_{min}$,  where $y_{max}$ and $y_{min}$ are the maximum and minimum labels in the dataset, respectively, and
>
> (5) $\mathcal{H}$ is the finite hypothesis space of prediction models.
>
> We can see that if one directly uses the original label distribution in the training objective function, i.e., $\hat{P}_{y}=P_y$:
>
> (a) The "bias" term will be $0$.
>
> (b) However, the "variance" term will be extremely large for minority data because $\hat{P}_{y}$ is very close to $0$.
>
> In contrast, under N.I.D., $\hat{P}_{y}$ used in the training objective function will be smoothed. Therefore:
>
> (a) The minority data's label density $\hat{P}_{y}$ will be smoothed out by its neighbors and becomes larger (compared to the original $P_y$), leading to smaller "variance" in the generalization error bound.
>
> (b) Note that $\hat{P}_{y}\neq P_y$, VIR (with N.I.D.) essentially increases bias, but significantly reduces its variance in the imbalanced setting, thereby leading to a lower generalization error.
>
> We will include the discussion above and our proof in the revision as suggested.
>
> [1] Delving into Deep Imbalanced Regression.
>
> [2] Learning multiple layers of features from tiny images.
>
> [3] Deep Expectation of Real and Apparent Age from a Single Image Without Facial Landmarks.

---

> > ### Comment · Reviewer_ivhk · 2023-08-15
> >
> > Thanks for the detailed response.
> >
> > I have read the Global Response for Q1, but I do not figure out why you partition the label space into B equal-interval bins, can you make a detailed explanation?

---

> > > ### Author Response · Authors · 2023-08-15
> > > **Thank You for Your Further Response**
> > >
> > > Thank you for your further response. This is a good question.
> > >
> > > **Why We Need Bins.** Throughout our method, we need to compute the statistics (i.e., the mean and variance) and the "statistics of statistics" of data points (Line 164-165); computing these statistics (e.g., the mean) requires a group of data points. Therefore, we need to partition the continuous label size into $\mathcal{B}$ bins. For example, in the equations from Line 176-177, e.g., $\mu_b^{\mu} = \frac{1}{N_b} \sum\nolimits^{N_b}_{i=1} z_i^{\mu}$, we need to compute the statistics of bin $b$, which contains $N_b$ data points in the bin. As mentioned by Reviewer 9XGJ, it is ``"common"`` to ``"use the quantization- (= binning-) based approach to regression problems"``.
> > >
> > > It is also worth noting in the extreme case where (i) each data point has a different label $y$ and (ii) we use a very small bin size, each bin will then contain exactly only one data point.
> > >
> > > **Equal-Interval Bins versus Equal-Size Bins.** Note that since our smoothing kernel function is based on labels (i.e., $k(y, y')$), it is more reasonable to use **equal-interval** bins rather than **equal-size** bins.
> > >
> > > **(1)** For example, if we use the equal-interval bins $[0,1),[1,2),...$, VIR will naturally compute $k(y, y')$ for $y=1,2,3,4,5,...$ and $y'=1,2,3,4,5,...$.
> > >
> > > **(2)** In contrast, if we use equal-size bins, VIR may end up with *large intervals* and may lead to inaccurate kernel values for $k(y, y')$. To see this, consider a case where equal-size bins are $[0,1),[1,2),[2,3.1),[3.1,8.9),...$; the kernel value $k(y, y')$ between bins $[2,3.1)$ and $[3.1,8.9)$ is $k(2,3.1)$, which is very inaccurate since $3.1$ is very far away from the mean of the bin $[3.1,8.9)$ (i.e., $6$). Using small and equal-interval bins can naturally address such issues.
> > >
> > > Thank you again for keeping the communication channel open, and we will be very happy to provide more details if you have any further questions.

---

> > > > ### Comment · Reviewer_ivhk · 2023-08-16
> > > >
> > > > Thanks for your explanation, my concerns are now addressed.
> > > >
> > > > I will increase my score by 1 for your detailed and convincing response.

---

> > > > > ### Author Response · Authors · 2023-08-16
> > > > > **Thank You**
> > > > >
> > > > > Thank you for your prompt and encouraging response as well as for increasing the score! We are glad that our response has been helpful and convincing.

---

### Official Review · Reviewer_cFEB · 2023-07-07

**Soundness:** 3 good
**Presentation:** 3 good
**Contribution:** 3 good
**Rating:** 6
**Confidence:** 3

**Summary:**

In this work, authors recognize that although the existing regression models for the imbalanced datasets have been mainly developed to improve the prediction accuracy, they overlooked the quality of the uncertainty estimation. In this context, authors propose a deep probabilistic regression framework, to improve the uncertainty estimation performance as well by combining the idea of [1] and [2].

Specifically, authors first consider multiple bins, splitted on the range of labels to get statistic of the labels. Then, they revise the latent statistic of VAE for the imbalanced datasets, by smoothing these features based on the statistic on each bins and then applying probabilistic whitening and recoloring procedure. Next, they use the revised latent features to get the posterior distribution of NIG distribution, which acts as the pseudo counts that alleviates the issue of imbalanced sets. Last, they employ these parameters for prediction and training.

Empirically, authors demonstrate that the proposed approach can improve the performance of the prediction accuracy and uncertainty estimation on various datasets.

[1] Delving into Deep Imbalanced Regression - ICML 21

[2] Deep Evidential Regression - NeurIPS 20

**Strengths:**

* This work extends the DIR of [1], as the probabilistic model, to estimate the uncertainty.

* This work considers to use the NIG posterior distribution, updated by the statistic of the stochastic latent feature. This seems to help balance the latent features of the imbalanced labels and thus yield the credible uncertainty for the imbalanced datasets. I believe that this is a novel part of this work as comparing [1] and [2].

[1] Delving into Deep Imbalanced Regression - ICML 21

[2] Deep Evidential Regression - NeurIPS 20

**Weaknesses:**

* Absence of ablation study
> In experiment section, it seems that the proposed method improves the prediction accuracy and uncertainty estimation. However, the current work does not investigate (1) whether each trick of the proposed method, such as the use of the stochastic latent feature (VAE) and use of the posterior distribution (NIG), is effective and (2) whether the proposed method is consistent up to the number of bins.

* Less explanation on why the evidential regression model is used along with DIR, instead of using other BNNs.
> In general, BNNs is widely used to estimate the uncertainty. I believe that applying the BNNs with DIR could be a direct way to solve the targeted problem. However, author takes the evidential regression approach, without explaining the reason or its motivation. This seems to less persuade why the proposed method is reasonable. If authors provide the motivation of this approach or can demonstrate that the proposed method could outperform the results of the DIR, obtained by BNNs or other approach for uncertainty estimation, I believe that the contribution of this work would be more clear and strong.

**Questions:**

* Is the performance of the proposed method consistent up to the number of bins? How can performance vary with the number of bins?

* Why did the authors employ the stochastic latent feature through VAE, that is different approach to the [1] ?
 Does the stochastic feature improve the performance as comparing when the deterministic statistic is used to update the parameters of the NIG posterior distribution ?

* Did the authors consider the BNNs with DIR or the Deep ensemble approach [2] with DIR to solve the imbalanced regression problem ?


[1] Delving into Deep Imbalanced Regression - ICML 21

[2] Simple and Scalable Predictive Uncertainty Estimation using Deep Ensembles - NeurIPS 17

**Limitations:**

See above Weaknesses and Questions.

---

> ### Author Rebuttal · Authors · 2023-08-07
>
> Thank you for your constructive comments and insightful questions. We are glad that you find our work ``"novel"``. Below we address your questions one by one.
>
> **Q1.1: "...ablation study...not investigate (1) whether each trick ... the stochastic latent feature (VAE) ... the posterior distribution (NIG), is effective ..."**
>
> Thank you mentioning this. Actually we did perform ablation studies and the results have been included in the Supplementary Material (please see Tables 8 and 9).
>
> Besides, according to your suggestions, we have also performed additional ablation studies. The results in Tables 3.1 through 3.2 below (more results in **Table 1 of the one-page PDF file**) demonstrate the effectiveness of each element in our proposed method. **VIR w/o VAE** refers to VIR *without stochastic latent features*, i.e., "the deterministic statistic is used to update the parameters of the NIG posterior distribution". **VIR w/o NIG** refers to VIR without NIG. Note that **VIR w/o VAE & NIG** is equivalent to **LDS+FDS+DER** in our main paper; here we include its results in the table as a reference to further demonstrate the effectiveness of our method.
>
> Table 3.1: Ablation studies on AgeDB in terms of MSE
>
> | model | overall | many | median | few |
> | :---------: | :------: | :------: | :------: | :------: |
> |VIR w/o VAE & NIG| 112.62 | 94.21 | 140.03 | 210.72 |
> |VIR w/o NIG| 87.48 | 73.72 | 107.64 | 161.69 |
> |VIR w/o VAE| 96.46 | 86.72 | 102.56 | 171.52 |
> |VIR (Ours)| **81.76** | **70.61** | **91.47** | **142.36** |
> |||||
>
> Table 3.2: Ablation studies on AgeDB in terms of NLL
>
> | model | overall | many | median | few |
> | :---------: | :------: | :------: | :------: | :------: |
> |VIR w/o VAE & NIG| 3.787 | 3.689 | 3.912 | 4.234 |
> |VIR w/o NIG| 3.722 | 3.604 | 3.821 | 4.209 |
> |VIR w/o VAE| 3.784 | 3.685 | 3.866 | 4.218 |
> |VIR (Ours)| **3.703** | **3.598** | **3.805** | **4.196** |
> |||||
>
> It's worth noting that we included similar ablation studies in Tables 8 and 9 of the Supplementary Material, where the "Encoder-only VIR" is equivalent to "VIR w/o NIG & LDS", and "Predictor-only VIR" is equivalent to "VIR w/o VAE & LDS". These results also verify the effectiveness of each element in our proposed method.
>
> we will incorporate the additional ablation studies above into the Supplementary Materials in the revision as suggested.
>
> **Q1.2: "...  method is consistent up to the number of bins." " performance vary with the number of bins?"**
>
> This is a good question. Please refer to **Q1 in the Global Response**.
>
> **Q2: "... why the evidential regression model is used along with DIR, instead of using other BNNs."**
>
> This is a good point. We do not consider other BNNs in this work because:
>
> **(1)**  **Weights** in Bayesian Neural Networks (BNNs) are extremly high-dimensional; therefore BNNs have several limitations, including the intractability of directly inferring the posterior distribution of the **weights** given data, the requirement and computational expense of sampling during inference, and the question of how to choose a **weight** prior [3]. In contrast, evidential regression does not have these challenges.
>
> **(2)** In our preliminary experiments, we found that typical BNN methods suffer from computational inefficiency and would require at least two to three times more computational time and memory usage. In contrast, evidential regression does not involve such computation and memory overhead; its overhead only involves the last (few) layers, and is therefore minimal.
>
> **(3)** Additionally, as demonstrated in [2], Deep Ensemble typically performs as well as or even better than BNNs. Our method outperforms **Deep Ensemble** (Tables 1~6 in the paper, with more results in **Q4** below), therefore suggesting its superiority over typical BNN methods.
>
> We appreciate your suggestion and will incorporate the discussion above into our revised paper.
>
> **Q3: "Why ... employ the stochastic latent feature through VAE ...? Does the stochastic feature improve ...?"**
>
> We are sorry for the confusion. Please refer to **Q2 in the Global Response**.
>
> **Q4: "... consider the BNNs with DIR or the Deep ensemble [2] with DIR ...?"**
>
> As mentioned **Q2**'s response, we did not include other BNNs due to their limitations in the context of our work. Furthermore, as demonstrated in [2], Deep Ensemble typically performs as well as or even better than BNNs. Our method outperforms Deep Ensemble (Tables 1~6 in the paper), therefore suggesting its superiority over typical BNN methods.
>
> According to your suggestion, we ran additional experiments on **combining Deep Ensemble and DIR** and report the results in Tables 4.1~4.2 below (more results in **Table 3 of the one-page PDF file**). These results show that while this combination allows DIR to produce uncertainties, our method can still outperform it by a large margin.
>
> Thanks for your insightful question, and we will include these additional results and the discussion above in the revision.
>
> Table 4.1: Results for DIR + Deep Ensemble in terms of MSE
>
> | model | overall | many | median | few |
> | :---------: | :------: | :------: | :------: | :------: |
> |DIR + Deep Ensemble| 94.10 | 80.24 | 109.45 | 182.52 |
> |VIR (Ours)| **81.76** | **70.61** | **91.47** | **142.36** |
> |||||
>
> Table 4.2: Results for DIR + Deep Ensemble in terms of NLL
>
> | model | overall | many | median | few |
> | :---------: | :------: | :------: | :------: | :------: |
> |DIR + Deep Ensemble| 5.069 | 4.772 | 4.574 | 5.236 |
> |VIR (Ours)| **3.703** | **3.598** | **3.805** | **4.196** |
> |||||
>
> [1] Delving into Deep Imbalanced Regression.
>
> [2] Simple and Scalable Predictive Uncertainty Estimation using Deep Ensembles.
>
> [3] Deep Evidential Regression.

---

### Author Rebuttal · Authors · 2023-08-07

We thank all reviewers for their encouraging and constructive comments. We are glad that they found the problems we identified ``"important"``/``"novel"`` (ivhk, 9XGJ, C6MR), our idea/method ``"novel"``/``"non-trivial"``/``"reasonable"`` (cFEB, C6MR, ivhk), our paper ``"easy to follow"``/``"clear"`` (9XGJ, C6MR), and that our method has ``"excellent"`` performance (9XGJ) and ``"superiority"`` (C6MR) over SOTA methods in terms of both ``"accuracy"`` and ``"uncertainty estimation"`` (C6MR, ivhk). Below we address the reviewers' questions one by one.

Due to the space constraint, i.e., 6000 characters per reviewer, we cannot cover every question in the rebuttal, but we promise to address all questions, cite all related references, and **include all discussions/results below in our revision**.

**[cFEB, ivhk] Q1. The Number of Bins.**

Our preliminary results indicate that the performance of our VIR (as compared to the SOTA baseline, i.e., Ranksim [2]) remains consistent regardless of the number of bins, as shown in the included tables below (more results in **Table 2 of the one-page PDF file**). Here we report results for the cases with $100/1=100$, $100/3\approx 33$, and $100/5=20$ bins.

Table 1: Ablation studies on the number of bins in terms of MSE

| model | bins | overall | many | median | few |
| :---------: | :------: | :------: | :------: | :------: | :------: |
| Ranksim | 100 | 83.51 | 71.99 | 99.14 | 149.05 |
| VIR (Ours) | 100 | **81.76** | **70.61** | **91.47** | **142.36** |
| Ranksim | 33 | 109.45 | 91.78 | 128.10 | 187.13 |
| VIR (Ours) | 33 | **84.77** | **77.29** | **95.66** | **125.33** |
| Ranksim | 20 | 98.71 | 84.38 | 107.89 | 171.04 |
| VIR (Ours) | 20 | **84.05** | **72.12** | **100.49** | **151.25** |
||||||

In our paper, we chose to use the same number of bins as the imbalanced regression literature [1, 2] for fair comparison with prior work. For example, in the AgeDB dataset where the regression labels are people's "age" in the range of 0~99, we use 100 bins, with each year as one bin. We will include the discussion and results above in the revised paper.

**[cFEB, ivhk] Q2: Advantage of Stochastic Latent Features / Dual Representation.**

That is a good question. Yes, stochastic latent features (i.e., dual representation) indeed improve the performance. We did conduct ablation studies in Table 8 and Table 9 of the Supplementary Material to verify this claim. In these tables, "Predictor-only VIR" is equivalent to our VIR without stochastic latent features and LDS. These results verify the effectiveness of such stochastic latent features.

We have also conducted additional ablation studies to further support our claim. The results in Tables 2.1 and 2.2 below (more results in **Table 1 of the one-page PDF file**) demonstrate the importance of stochastic latent features (i.e., dual representation) in our proposed method.

In the tables, **VIR w/o VAE** refers to VIR without stochastic latent features. These results verify that the stochastic latent features can indeed improve the performance.

Table 2.1: Ablation studies on AgeDB in terms of MSE

| model | overall | many | median | few |
| :---------: | :------: | :------: | :------: | :------: |
|VIR w/o VAE| 96.46 | 86.72 | 102.56 | 171.52 |
|VIR (Ours)| **81.76** | **70.61** | **91.47** | **142.36** |
|||||

Table 2.2: Ablation studies on AgeDB in terms of NLL

| model | overall | many | median | few |
| :---------: | :------: | :------: | :------: | :------: |
|VIR w/o VAE| 3.784 | 3.685 | 3.866 | 4.218 |
|VIR (Ours)| **3.703** | **3.598** | **3.805** | **4.196** |
|||||

**[ivhk, 9XGJ] Q3: Why Eqn. 5 Focuses on Minority Data (Line 42-43) and Alleviates the Imbalance Problem.**

We are sorry for the confusion. To see why, note that Eqn. 5 is the *negative log likelihood* for the Normal Inverse Gaussian (NIG) distribution.

Specifically, each posterior parameter ($\nu_i^*, \gamma_i^*,  \alpha_i^*$) of the NIG distribution is reweighted by importance weights, thereby assigning higher weights to minority data during training and allowing minority data points to benefit more from their neighboring information.

Take $\nu_i^*$ as an example. Assume a minority data point $(x_i,y_i)$ that belongs to bin $b$, i.e., its label $y_i=y_b$. Note that there is **a loss term** $(y_i-\gamma_i^*)^2\nu_i^*$ in **Eqn. 5**, where $\gamma_i^*$ is the model prediction, $y_i$ is the label, and $\nu^{*}_{i}$ is the "importance weight" for this data point.

Here $\nu_i^* = \nu_0 + (\sum_{b' \in \mathcal{B}} k (y_b, y_{b'}) p(y_{b'}))^{-1/2} \cdot n_i$ where $n_i$ represents the pseudo-count for the NIG distribution. Since $(x_i,y_i)$ is a minority data point, data from its neighboring bins has smaller frequency $p(y_{b'})$ and therefore smaller $\sum_{b' \in \mathcal{B}} k (y_{b}, y_{b'}) p(y_{b'})$, leading to **a larger "importance weight"** $\nu_i^*$ **for this minority data point in Eqn. 5**.
This "allows the negative log likelihood (Eqn. 5) to naturally put more focus on the minority data" (Line 42-43), thereby alleviating the imbalance problem.

**[ivhk] Q4: The Choice of the Kernel Functions.**

The DIR paper [1] shows that a simple Gaussian kernel with inverse square-root weighting (i.e., SQINV) achieves the best performance. We therefore use SQINV thoughout our paper for fair comparison (more details in Line 307 of the main paper).

Besides, our preliminary results also show that the performance is not very sensitive to the choice of kernels, as long as the kernel $k(a,b)$ reflects the distance between $a$ and $b$, i.e., larger distance between $a$ and $b$ leads to smaller $k(a,b)$.

[1] Delving into Deep Imbalanced Regression.

[2] RankSim: Ranking Similarity Regularization for Deep Imbalanced Regression.

---

### Decision · Program_Chairs · 2023-09-21

**Decision:**

Accept (poster)

**Comment:**

The paper proposes a varaitional (Bayesian variant of) imbalanced regression, based on the neighboring and identically distributed assumption.   Experiments show sota results in terms of accuracy and uncertainty estimation.

Reviewers acknowledged strengths,

- Novel idea for an important and little-explored topic
- Assumption reasonable
- Good performance in both accuracy and uncertainty estimation, outperforming baselines.
- Well written,

while they raised several concerns,

- Ablation study missing? (actually in Appendix)
- Comparison with BNN, instead of DIR, missing.
- Clarity issues
- Subjective statements
- Code is not available.
- Limited novelty (combination of sota)

The authors' rebuttal addressed all reviewer's concerns, and all reviewers recommend acceptance.

Typos should be fixed.  For example, LHS and RHS don't match in the equation in Line 112.